# Characterization of RNA polymerase II trigger loop mutations using molecular dynamics simulations and machine learning

**Bercem Dutagaci[1]\*, Bingbing Duan[2], Chenxi Qiu[3], Craig D. Kaplan[2], Michael Feig[4]\***

**1** Department of Molecular and Cell Biology, University of California Merced, Merced, California, United States of America, **2** Department of Biological Sciences, University of Pittsburgh, Pittsburgh, Pennsylvania, United States of America, **3** Department of Genetics, Harvard Medical School, Boston, Massachusetts, United States of America, **4** Department of Biochemistry and Molecular Biology, Michigan State University, East Lansing, Michigan, United States of America

\* bercemdutagaci@gmail.com (BD); mfeiglab@gmail.com (MF)

**Data Availability Statement:** The experimental data is in S1 Spreadsheet, average distances extracted from MD are in S2 Spreadsheet. The ML models and protocols are found at: https://github.com/bercemd/PolII-mutants All derivative analysis

## Abstract

Catalysis and fidelity of multisubunit RNA polymerases rely on a highly conserved active site domain called the trigger loop (TL), which achieves roles in transcription through conformational changes and interaction with NTP substrates. The mutations of TL residues cause distinct effects on catalysis including hypo- and hyperactivity and altered fidelity. We applied molecular dynamics simulation (MD) and machine learning (ML) techniques to characterize TL mutations in the *Saccharomyces cerevisiae* RNA Polymerase II (Pol II) system. We did so to determine relationships between individual mutations and phenotypes and to associate phenotypes with MD simulated structural alterations. Using fitness values of mutants under various stress conditions, we modeled phenotypes along a spectrum of continual values. We found that ML could predict the phenotypes with 0.68 $R^2$ correlation from amino acid sequences alone. It was more difficult to incorporate MD data to improve predictions from machine learning, presumably because MD data is too noisy and possibly incomplete to directly infer functional phenotypes. However, a variational auto-encoder model based on the MD data allowed the clustering of mutants with different phenotypes based on structural details. Overall, we found that a subset of loss-of-function (LOF) and lethal mutations tended to increase distances of TL residues to the NTP substrate, while another subset of LOF and lethal substitutions tended to confer an increase in distances between TL and bridge helix (BH). In contrast, some of the gain-of-function (GOF) mutants appear to cause disruption of hydrophobic contacts among TL and nearby helices.

## Author summary

RNA polymerase II (Pol II) synthesizes RNA with the help of an active site domain called the trigger loop (TL). Mutations in the TL cause changes in the activity of Pol II that range from gain-of-function (GOF, viable but hyperactive) to loss-of-function (LOF, viable but hypoactive) or lethal. This study provides a systematic characterization of the structural

data is presented in Figs 1–6 and S1–S26, and S1 and S2 Tables. Code is made available via github: https://github.com/bercemd/PolII-mutants.

**Funding:** This study was funded by the National Institutes of Health (R35 GM126948, to MF, R01 GM097260 and R35 GM144116 to CDK). Computer time at XSEDE facilities was used under grant TG-MCB090003. The funders had no role in study design, data collection and analysis, decision to publish, or preparation of the manuscript.

**Competing interests:** The authors have no competing interests to disclose.

and functional outcomes of the TL mutations using molecular dynamics (MD) simulations and machine learning (ML). We obtained functional phenotypes of mutants by ML using genetic fitness scores (measure of growth defect strength) as input. We revealed that mutant TL sequences could predict the functional outcomes at a relatively high correlation. Then, we performed MD simulations to relate structural information to the phenotypes. The analysis of the MD data suggested that there are two subsets of lethal and LOF mutants, where one subset had increased distances between the TL and the substrate, while the other subset showed increased distances between TL and another active site domain called the bridge helix (BH). On the other hand, some of the GOF mutants altered a key hydrophobic pocket formed by interactions between residues near the active site. Overall, this study enhances our understanding of the effects of TL mutations to the Pol II function.

## Introduction

RNA polymerase II (Pol II) is the enzyme that synthesizes mRNA in eukaryotes. Structural [1,2,3,4] and computational [5,6,7,8] studies have provided insights into the mechanism of this process, which takes place by repeating a nucleotide addition cycle (NAC) for addition of new nucleotides to the nascent RNA [9]. Proposed mechanisms for the NAC emphasize conformational changes of a highly conserved domains in the active site, present within the largest subunit of yeast Pol II, Rpb1, but the insights gained here can be transferred to other polymerase systems due to a high level of conservation of active site domains. One of these domains is called the trigger loop (TL) and the other is the nearby bridge helix (BH). The TL has open and closed conformations, which are known to be important for nucleotide addition [10,11,12,13]. The NAC starts with the Pol II complex with an open TL that allows an incoming nucleoside triphosphate (NTP) to enter the active site. Upon initial binding of the NTP, the TL closes and catalysis is promoted for substrates base-paired with the template. This results in the pre-translocation (substrate added) state, followed by TL opening together with pyrophosphate ion (PPi) release, and subsequent or concurrent translocation [14]. The TL has been suggested to have an important function in selecting [4,15,16,17] and positioning [18,19] the correct NTP at the active site and in affecting the kinetics [15,20,21] of the NAC. TL involvement during transcriptional pausing [22,23], backtracking [24,25] and translocation [10,26,27] has also been proposed.

Detailed mechanisms of TL function are still not fully known. Previous studies suggest that TL is crucial for transcription by showing that a complete deletion of the TL from different species caused marked reductions in transcription rate [23,28,29] In the case of deletion of TL, transcription could still take place but with a large decrease, $10^2$–$10^4$ fold, in the catalysis rate, [28,29] and with a significant compromise in fidelity [29]. These studies suggest that TL is playing a fundamental role in transcription. Certain residues were identified to be especially important for function, such as H1085, L1081, E1103 and Q1078 [4,15,16,18,26,30,31,32]. H1085 and L1081 are in close distance to the NTP when the TL is closed. Therefore, their roles have been attributed to the positioning of the correct NTP. Most of substitutions of H1085 and L1081 are lethal [4,18,32]. On the other hand, E1103 mutations are known to cause an increased catalytic rate but with compromised fidelity [15,16,26,32]. Further studies showed that Q1078 has interactions with the sugar moiety of the NTP, and most of its mutations are also not viable [30,31,32]. Because the TL must support multiple conformations, there may be complex effects of specific substitutions. Site-directed mutagenesis and a prior comprehensive

genetic study of TL alleles together suggest Pol II mutant phenotypic classes and complex interactions between residues supporting a functional network [21,32]. In that study, the effects of mutations were classified broadly as either 'loss of function' (LOF), where catalytic activity is, or is predicted to be, reduced *in vitro*, 'gain of function' (GOF), where catalytic activity is, or is predicted to be, increased, or 'lethal', where essential functions are compromised. As a result, genetic phenotypes were associated different functional outcomes providing a framework for insights into the role of different TL residues during transcription.

The effects of mutations on proteins have been studied previously by computational methods. Many studies report MD simulations that predict structural effects of mutations and link those effects with function [18,33,34]. These studies have brought invaluable insights into the effects of mutations but covering a large number of mutations with MD simulations is computationally challenging. Recently, machine learning approaches have become widely used for predicting the effects of mutations on various properties like protein stabilities [35,36], ligand binding [37,38,39], variant fitness [40,41] and functional phenotypes [42,43,44]. In these studies, the input typically consists of amino acid sequences, evolutionary data, structural information and biochemical data. A range of machine learning approaches have been applied, including feed-forward neural networks [39,41,42], variational autoencoder (VAE) models [43,44], convolutional neural networks [36,39,40], and ensemble learning methods [35,37,38,41]. The predictive performance measured in terms of Pearson coefficients typically ranges from 0.5 to 0.8 [35,37,38] suggesting that machine learning models can be useful in predicting effects of mutations. Here, we apply similar machine learning frameworks to predict functional outcomes for Pol II. Different from previous studies, we added input from MD simulations as additional features with the goal of gaining additional insights by combining both approaches to predicting function.

How the functional TL phenotypes that result from residue variations are manifested is an open question, since structural and dynamic details at the atomic level for individual mutants is lacking, as is understanding of potential commonalities at the biochemical level within mutant classes. To address this, we combined the data from experimental fitness scores and molecular dynamics (MD) simulations for TLs with different amino acid sequences to predict functional and structural outcomes of TL mutations using machine learning (ML) frameworks. The analysis here is based on an updated fitness dataset that extends the earlier analysis of Qiu *et al*. to develop a complete TL mutation phenotype map based on a continuous representation of functional phenotypes [32]. First, we developed ML models using amino acid sequences to predict TL mutation phenotypes. Then, we selected 135 TL single mutants with known functional phenotypes and performed atomistic molecular dynamics (MD) simulations of those mutants. Following MD simulations, we applied ML algorithms on data extracted from the simulations to develop a better understanding how different phenotypes map onto differences in structure and dynamics of Pol II near the active site. The structural data obtained from the MD simulations was primarily used to provide a mechanistic understanding of the TL mutant phenotypes when used in a VAE framework that allowed us to map function to structural features. Specific insights from this analysis are that lethal and LOF mutants have increased intramolecular distances between TL residues and the NTP for one subset of mutants, while for another subset of the lethal and LOF mutants, large distances between TL and BH residues are observed. We also predicted two distinct classes of GOF phenotypes where both affect a hydrophobic pocket formed by active site residues while a subset has increased BH-TL interactions. Overall, these findings lead to further understanding of the specific roles of the TL and the BH during Pol II function. This study also suggests that longer MD simulations, on possibly μs time scale, might be required to enhance the inference of the mutant mechanisms.

## Results

This study focuses on the interpretation of Pol II TL mutation phenotypes based on experiments via MD simulations and ML to develop a deeper mechanistic understanding of the role of the TL during transcription. We describe here three sets of results: 1) Based on fitness data from second-generation deep mutational scanning of the Pol II TL in *S. cerevisiae* (see methods), we generated a model for the classification of TL mutants along a continuous phenotypic spectrum and projected continuous phenotypes on a reduced dimensional latent space generated using a VAE model; 2) we applied ML to infer TL mutant phenotypes from TL sequence with and without structural data from MD simulations using a subset of gold standard mutants as training data; and, 3) we extracted mechanistic principles for how different classes of TL mutations modulate Pol II function based on VAE models that were trained on MD simulation data.

### Continuous Pol II function phenotypes for TL mutations from experimental fitness data

Pol II TL is highly conserved among three domains of life (Fig 1A) especially for the residues that are close to the catalytic site (Fig 1B). The level of conservation can be correlated with the importance of each residue for function. T1077 to G1088 are overall the most conserved residues of TL (Fig 1A), suggesting that any mutation on these residues may have a high impact on the function of the enzyme. Previous deep mutational scanning of the Pol II TL in yeast had classified mutants as lethal, LOF, GOF, or indeterminate [32]. In this study, we classified mutants based on a phenotypic numerical continuum instead of discrete classes. GOF, LOF, and lethal phenotypes were mapped to values of +1, -1 and -2, respectively and a value of 0 corresponds to the WT or indeterminate phenotypes those did not show any strong phenotype under any condition. A continuous phenotype better reflects gradual variations in functional outcomes. It also allowed us to project the phenotypes into a two-dimensional latent space and quantitatively analyze the transition between phenotypes. To place individual mutants along a phenotypic continuum, we first obtained quantitative fitness data for all possible TL single substitution mutants and a range of selected double mutants for a number of selection conditions (Fig 1C), (see Methods). These conditions have previously been shown to enable discrimination among hypo- and hyperactive Pol II mutants (LOF and GOF, respectively) [32]. We trained a neural network model (three hidden layers with 256, 128 and 64 nodes) based on the fitness data against the annotated functional phenotypes, converted to their corresponding numerical values, for a subset of the mutations described in the previous study [3,21,32]. The complete fitness set is provided in the S1 Spreadsheet and the training set contains 83 mutants annotated with their known phenotype classes. The resulting ML model was then used to predict the functional phenotype from the fitness data for the complete set of mutations in the TL as described in the Methods (Fig 1D). Most substitutions of residues between T1077 to G1088 were predicted as lethal or LOF consistent with the study of Qiu et al [32] and consistent with high sequence conservation for those residues (Fig 1A). These residues were also suggested to be important for function by earlier studies [4,12,18,19,29,45,46]. L1081 and H1085 are in close vicinity to the incoming NTP and they interact with the base and phosphate groups of NTP, respectively [4]. They were attributed to have functions in nucleotide selection, positioning and translocation [18,19,45,46]. In addition, F1084 and F1086 were suggested to have functions in stabilizing the orientation of H1085 that may play a role in NTP selection [12]. Q1078 is in close vicinity to the sugar of the incoming nucleotide [4] and was suggested to play a role in selecting the correct NTP [29]. Mutations of V1094, P1099, and to a lesser extent, L1105 broadly resulted in LOF phenotypes, suggesting importance for WT fitness. Conversely,

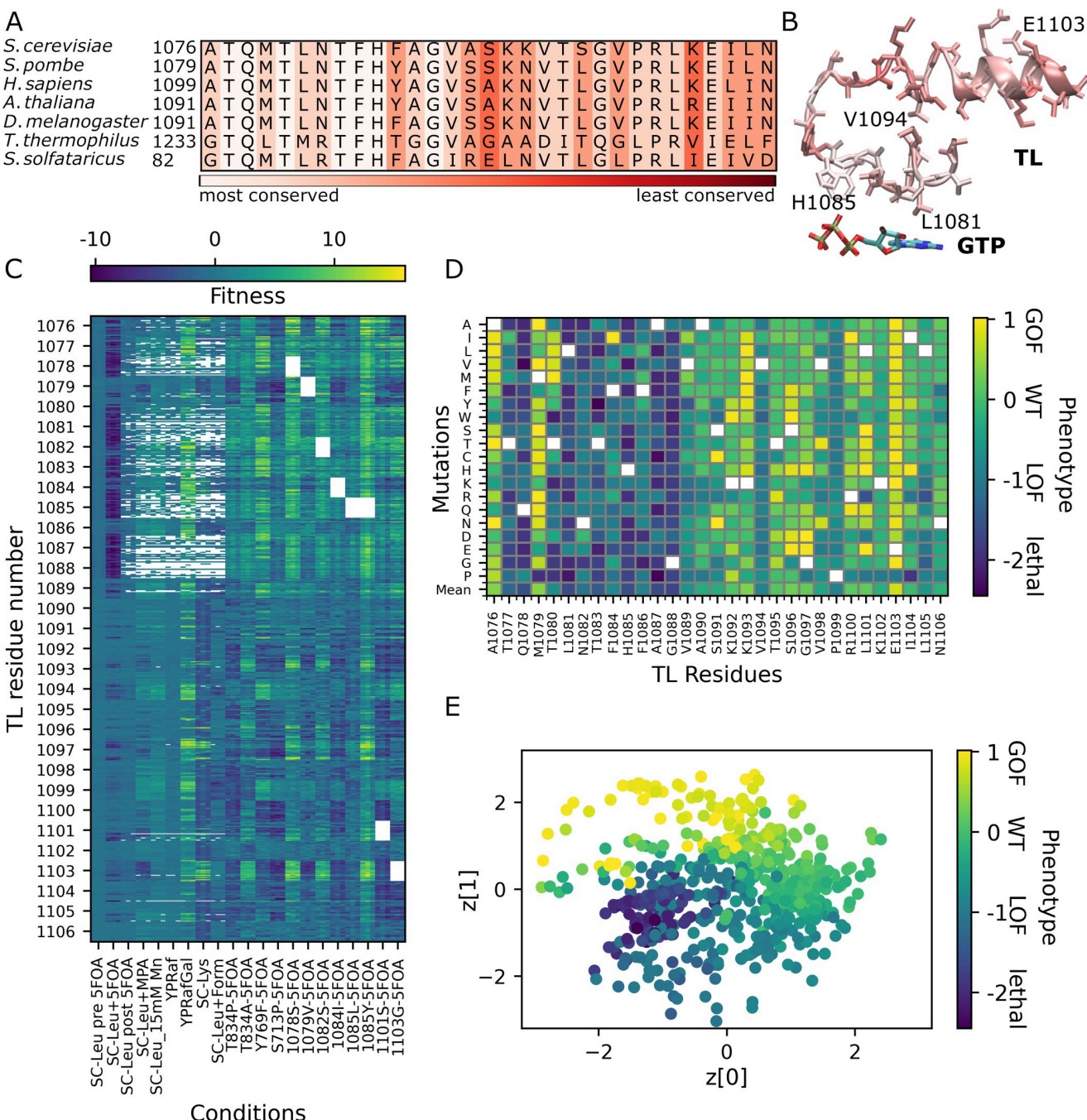

**Fig 1. TL sequence conservation and ML prediction of phenotypes from fitness data.** (A) Sequence conservation of TL in different organisms that are eukaryotes (*S.cerevisiae* and *S. pombe* from yeast, *H. sapiens* from mammals, *A. thaliana* from plants and *D. melanogaster* from insects), bacterium (*T. thermophilus*), and archaeon (*S. solfataricus*). (B) The structure of TL residues and GTP from *S.cerevisiae* (PDB code:2E2H). The TL is colored with the level of conservation among the species shown in A. (C) Fitness scores for mutants of TL residues under different conditions (see x-axis labels). The x-axis shows the different conditions with three replicates and the y-axis shows the TL mutations with 19 mutants for each residue, y-axis was labeled by only the TL residue number for avoiding a crowded labeling using all mutant names. For each residue, there are 19 mutants and for each mutant there are 21 data points with three replicates each. (D) Phenotypical landscape predicted from the fitness data. Each box is colored with the phenotype; white boxes reflect the WT amino acid at those positions; the row at the bottom depicted as "Mean" shows the average phenotypes for the residues. (E) Latent space of the unsupervised VAE model based on the fitness data with each data point colored according to its corresponding phenotype. The axes z[0] and z[1] are the first and second dimensions of the 2D latent space generated by projecting the fitness scores using the VAE model.

substitutions of selected residues mostly result in GOF phenotypes, as previously suggested. Among those, A1076, M1079, G1097, and L1101 form a hydrophobic pocket that may stabilize the open TL state [30]. Disruption of this pocket by mutations may facilitate TL closing leading to a GOF phenotype [30,32], albeit a bias towards TL closing may come at the expense of decreased fidelity [15,16,17,26,47,48]. In addition, selected mutations at K1092, K1093, R1100 and most at E1103 lead to GOF phenotypes. The residues K1092 and K1093 were suggested to stabilize the TL open state through interactions with other Rpb1 residues like D716, and D1309/E1280, respectively, shown by simulation and experimental studies [12,30,32]. Similarly, the GOF phenotype observed for E1103 substitutions was attributed to their stabilizing effect on the closed TL state [15,21,26].

To reduce dimensionality of the mutant space and associate the locations of mutants with different phenotypes, we mapped the fitness values of mutants onto a two-dimensional latent space by applying a VAE model (three layers with 256, 128 and 64 nodes in both encoder and decoder parts). There were two main reasons for using VAE model. First, VAE would provide clustering of similar phenotypes together in a reduced dimensional space without any supervision that would further support the accuracy of the predictions of continuous phenotypes obtained by a supervised model. The second reason is to use the reduce dimensional space to gather generalized information from the complex non-linear fitness data. The latent space captures the information in the fitness dataset in a reduced dimension, from which the fitness values can be regenerated with minimal loss as an output of the generative part of the model (decoder). Fig 1E shows the resulting latent space distribution of each mutant, colored according to the predicted phenotypes. We note that VAE models with 2D and 3D latent spaces provided similar generative performances (S1 Fig), therefore, we showed 2D model in Fig 1E. Although continuous phenotype predictions benefitted from supervision based on known phenotypes, the VAE model was trained without such supervision. Nevertheless, there is clear clustering of the mutants according to predicted phenotypes. The VAE model provides a regularized latent space with a more gradual transition of the phenotypes compared to the relatively more distinct locations of the different phenotypes in a 2D PCA analysis (S2 Fig). Moreover, the gradual transition between different phenotype classes suggests that this type of classification provides additional insights that are not captured by a discrete classification and that may be more consistent with an evolutionary fitness landscape. Transitions between phenotypes suggests that different phenotypes have similar fitness scores so that they are at the edge of their presented phenotypes and can be interconverted between phenotypes by additional mutations. Interestingly, transitions between nearby latent space projections are not just between neighboring phenotypes (*e.g.* from lethal to LOF and from LOF to neutral and then GOF) but also almost directly from GOF to lethal for some mutations (e.g. GOF mutants L1101E, G1097D, M1079A, E1103K, K1093F, F1084I) (S3 Fig). L1101E is at close distance in the latent space to LOF mutants A1087E and F1084Y and the lethal mutants G1088F and L1081S. The other GOF mutants, G1097D, M1079A, E1103K, K1093F, F1084I, are also at the boundary that they are close to LOF mutants F1084K, E1103P T1077D and the lethal mutant H1085F (S3 Fig). Their close distances to the lethal mutants on the latent space suggest that they have fitness values that are close to mutants causing serious defects on catalysis and, therefore, these GOF mutants are supposably most likely to be converted to lethality by additional mutations. An earlier study [21] on double mutations showed that the two GOF mutants at the boundary, which are G1097D and F1084I, turned to lethal upon an additional mutation, E1103G, which has GOF phenotype by itself. Although, E1103G is not at the border with the lethal mutants (S3 Fig), the additional effects of this mutation might have pushed the GOF mutants to the other side of the boundary.

## Inferring phenotypes from TL sequence

To further understand the information about fitness encoded in the TL sequence, we trained a supervised neural network (three layers with 128, 64, and 32 nodes and a flattening layer) to predict function from sequence. The target data were the continuous phenotype values determined from the experimental fitness data as described above (S1 Spreadsheet). In total, ten model replicates were generated for ten randomly generated training (100 mutants) and test (35 mutants) sets to result in 100 models in total. We note that decreasing the size of the training set to 80 reduces the performance of the models (average $R^2$ decreases from 0.52 to 0.44 for the test sets) while increasing it to 120 slightly increases the performance (average $R^2$ increases from 0.52 to 0.54 for the test sets) but limits the number of mutants in the test sets. Then, we took the models that provided the best $R^2$ and slope combination for each test set. The predictions from the best models for the ten sets were ensemble averaged to obtain the overall predictions. These are provided in the S1 Spreadsheet. The training and test loss of the ten models is shown in S4 Fig and the correlations for the training and test sets for each model are shown in S5 and S6 Figs, respectively. Fig 2A shows the average prediction performance with good correlation ($R^2 = 0.68$). However, the slope of 0.60 and a more limited range in predicted values compared to the actual phenotypes indicates that extreme outcomes (gain of function or lethal) are not predicted as reliably as the overall trend. This is also evident in the difference map for predictions shown in S7 Fig. The model also could not predict LOF phenotypes of specific substitutions in residues that otherwise have predominantly GOF mutants like in the case of E1103P. Fig 2B shows sequence-based phenotype predictions of single mutations. The predictions largely agree with the phenotypes from the fitness values, but, again, with less variation in the predicted values towards the extremes. We also compared sequence-based model with a simple model that predicts the phenotypes from the average phenotypes for each mutant from the training sets used in the sequence models (see S8 Fig). Compared to the sequence-based model we found a lower overall correlation ($R^2 = 0.59$) but an improved slope (0.71). However, for the sequence model, there were difficulties in predicting phenotypes that have a large deviation from the average phenotype of each residue (see the outliers of M1079P, I1104P, L1101R, T1080M in S7 Fig). Overall, the improved correlation obtained by the

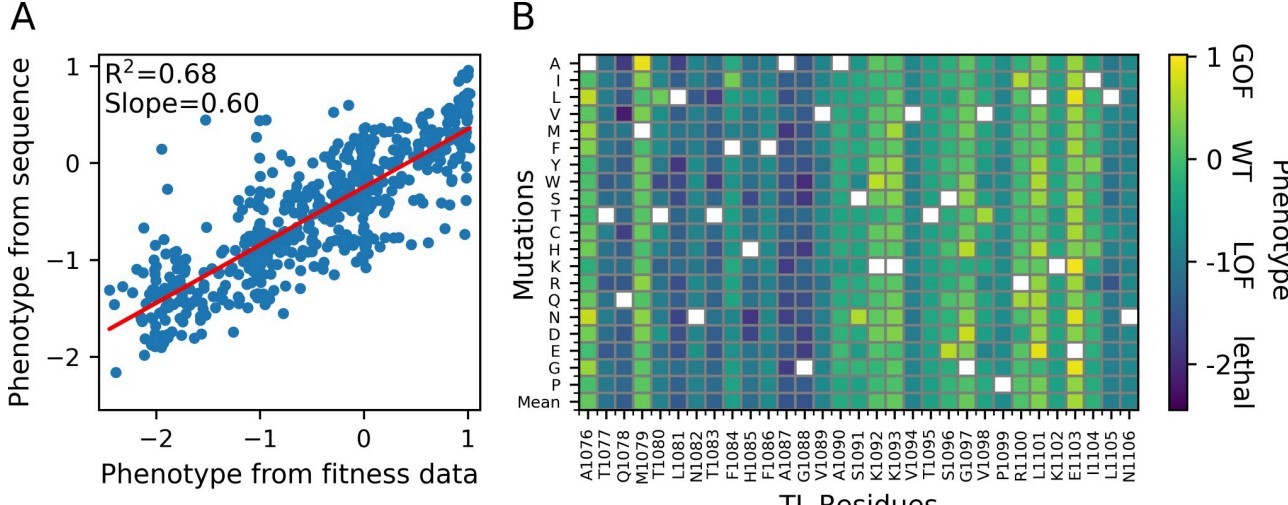

**Fig 2. Prediction of phenotype from TL sequence.** (A) Phenotype predicted from sequence vs. phenotype obtained from fitness data along with a linear regression curve. The predictions are the ensemble average values from models trained with 10 random training/test sets (B) Phenotypes predicted from the sequence for all single mutations.

sequence model over the simple model suggests that the TL sequence is a powerful predictive feature for inferring functional phenotypes of TL residue mutants.

We further tested the models trained on single mutants for the prediction of double mutants. Predictions of double mutants is practically more important since not all combinations of double mutants can be tested by experiments. Double mutants are interesting from a functional point of view to understand to what degree mutations have additive effects. At the same time, this analysis reveals limitations of a prediction model trained on single mutants for predicting the phenotypes of double mutants. We ensemble averaged the predictions of ten models as we did for the single mutants. Fig 3 shows the phenotypic landscapes of all individual TL substitutions combined with either E1103G (GOF), G1097D (GOF), F1084I (GOF) or Q1078S (LOF). Generally, our model predicted additive effects of double mutants on phenotypes in which similar phenotypes showed an increase effect while opposite phenotypes suppressed each other. More specifically, the GOF mutants (E1103G, G1097D, F1084I) were predicted to cause the suppression of LOF and lethal mutants across the entire set of additional mutations. To quantify the additivity of the double mutants, we calculated completely additive phenotypes by adding up single mutant phenotypes and showed that the predicted phenotypes are highly correlated with the additive phenotypes with $R^2$ of 1.0 and mean squared error of 0.29 (S9A Fig). S9B Fig showed that the predicted phenotypes are slightly higher than the additive ones. The agreement with additivity is higher for GOF-GOF mutants and tend to be correlated with the spatial distances between the mutation sites that the closer mutation sites

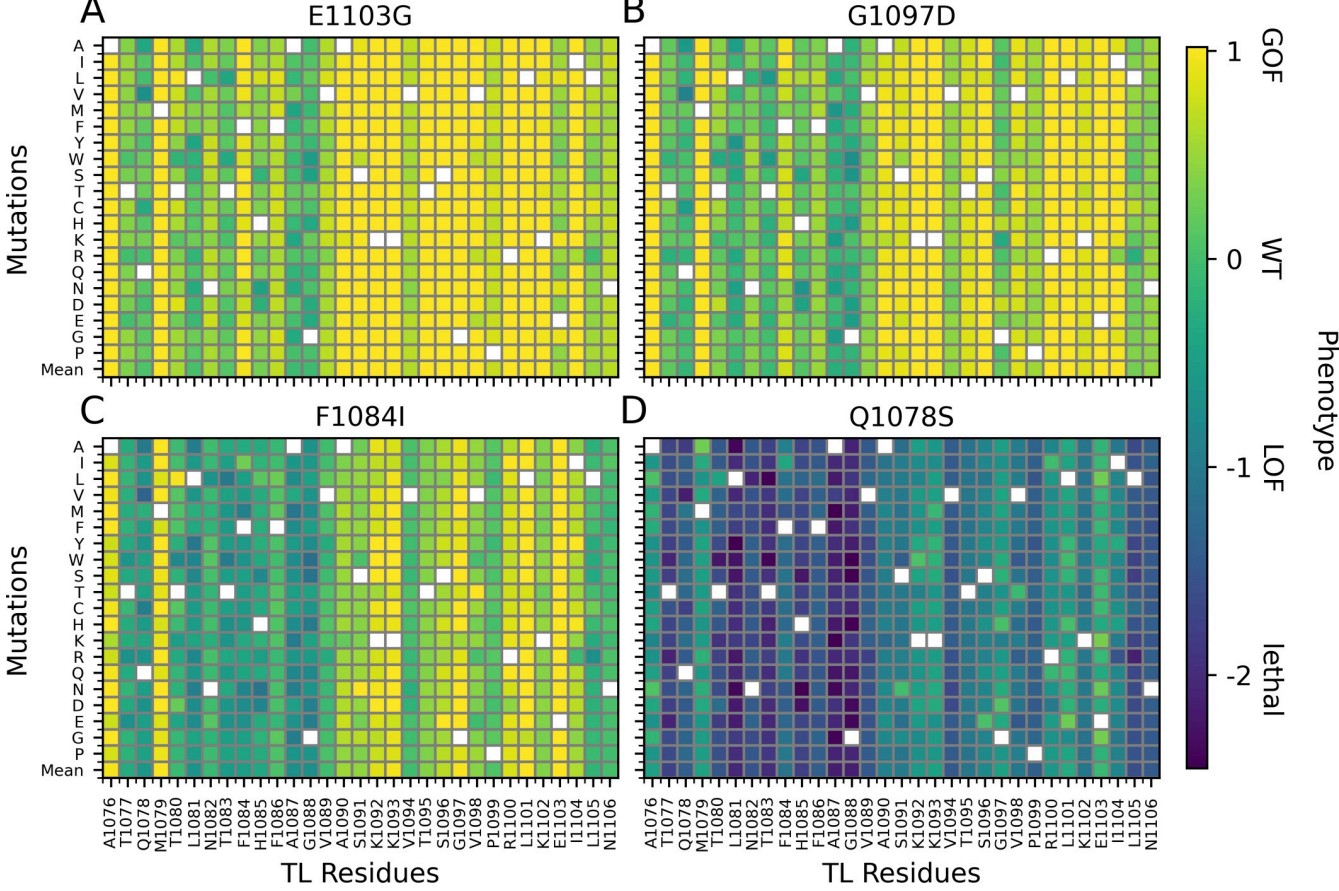

**Fig 3.** Prediction of phenotypes from TL sequence for double mutants of E1103G (A), G1097D (B), F1084I (C) and Q1078S (D).

provided a better agreement with the additive phenotypes. (S9B Fig). The overall additive prediction is consistent with previous studies for most of the cases [16,21]. For example, the combination of E1103G, which has a GOF phenotype, with LOF mutants F1086S, H1085Q and H1085Y resulted in enzyme activity between the two phenotypes, while combinations with lethal mutants of Q1078A, N1082A and H1085A resulted in suppression of lethality [21]. The predicted increased GOF phenotypes for GOF-GOF double mutants were in contrast to GOF combinations E1103G-G1097D and E1103G-F1084I having been found to be lethal [21]. These combinations were hypothesized to be lethal due to extreme GOF phenotypes crossing a threshold for viability where an overemphasized GOF phenotype may eventually disrupt the activity of the enzyme, an outcome not included in the training of ML model based on single mutations. Alternatively, a complete disruption of the hydrophobic pocket near the active site may result in larger structural changes that disable any polymerase activity. The LOF mutant of Q1078S also predicted additive effects by suppressing the GOF phenotypes and causing lethal or more severe LOF phenotypes for LOF-LOF double mutants. We note that neutral (WT) function may be restored for certain double mutations of M1079, K1092, K1093, G1097, L1101, when added to Q1078S according to the predictions. It remains to be tested experimentally whether such double mutants could in fact restore normal function.

## Inferring phenotypes from MD simulations of mutants

MD simulations were performed on 135 mutants to determine if emergent properties of the simulated mutants might provide better discrimination across mutant classes or between phenotypes, and, if so, what structural and dynamic properties might be hallmarks of specific phenotypical outcomes. Mutants were chosen based on the predictions from the previous studies [21,32]. To generate features for ML training, we extracted intramolecular distance data from the MD simulations (Fig 4). Specifically, we captured a subset of intramolecular distances from the MD trajectories deemed to be relevant for Pol II function and potentially sensitive to TL mutations: TL-TL residue pairs, TL-BH residue pairs and BH-BH residue pairs and TL residue-GTP pairs that are at close distance in the WT structure; GTP $P_\alpha$ and terminal RNA O3'distance relevant for catalysis; base pair distance between GTP-H1 and the corresponding DNA (18-DNA-N3) and the distance between sugar carbons of GTP-C1'and 18-DNA-C1'; finally, the distance between $Mg^{2+}$ and $P_\alpha$ of GTP. In total, 62 distances were calculated, and the distance lists and average distances are provided in the S2 Spreadsheet. We selected distances close to the TL since it has the mutation sites and distances close to the active site that are likely to be relevant to catalysis.

Fig 4A and 4B show the schematic representation of the distances and Fig 4C shows the free energy map for the phenotype vs. average distances for each mutant studied via simulation. Most of the distances stay between 2–12 Å, while as the phenotypes range from GOF (1.0) to lethal (-2), some distances become longer. However, the affected distances differ between mutants. The lethal mutants (phenotype < -1.5) mostly affect the GTP-TL distances since the heatmap shows minimum energies at large GTP-TL distances for the lethal mutants (S10 Fig). On the other hand, some of the LOF mutants seem to affect the TL-GTP, TL-TL and BH-TL pairs as they span larger distances at minimum energies. (S10 Fig). Overall, this suggests that lethal and LOF mutants cause increase in distances either of GTP or TL-BH residues that directly or indirectly impact the catalysis.

ML models (three layers with 128, 64 and 32 nodes) for predicting phenotypes were then trained using the MD data with and without sequence data (Figs 5 and S4–S6). ML models using only the MD-derived distances were clearly not as predictive as the models based on just amino acid sequence, even when a more complex model with an attention layer (two layers

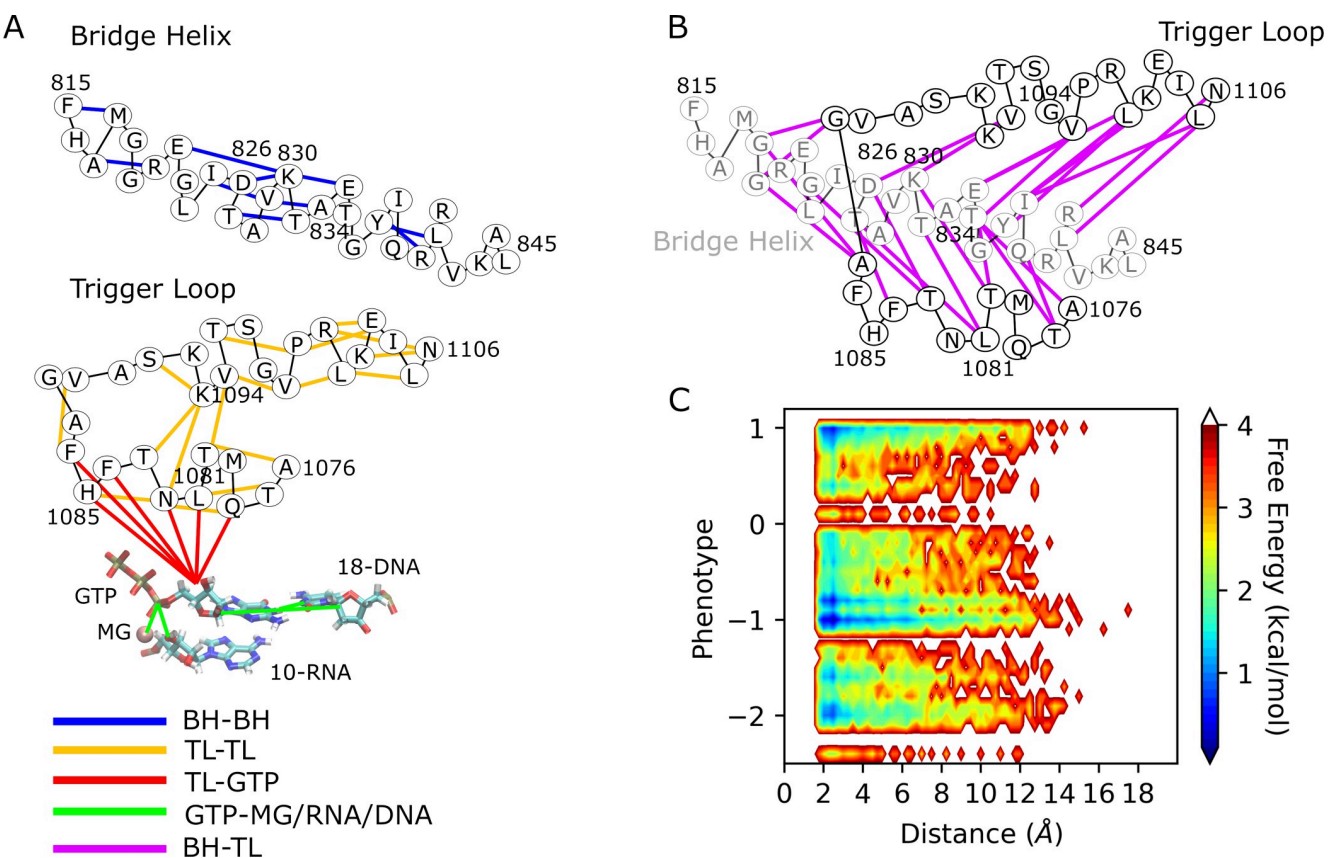

**Fig 4. Distance analysis of the MD trajectories.** (A) Schematic illustration of the pairs of BH-BH, TL-TL, TL-GTP and GTP with $Mg^{2+}$, terminal RNA and base-pair DNA and (B) BH-TL with the color codes given in the figure, the black lines are used to show the adjacent amino acids (C) heatmap plot of phenotypes vs average distances for the mutants from MD simulations. Distances are provided in the S2 Spreadsheet.

with 128, 64 nodes and an attention layer) was considered (Fig 5). We note that the $R^2$ values in Figs 2 and 5 are different since Fig 2 shows the correlations of the average predictions over the sets for all the mutants while Fig 5 shows the averaged $R^2$ over the sets for the mutants in the test sets. We also attempted to combine MD and sequence data and found that the MD data could not improve the predictions over using just the sequence information by itself. We interpret this finding to suggest that the MD data may be too noisy and sampling may be incomplete to reliably discern differences between specific mutations. We note that we used a different set of features which is backbone dihedral angles of TL residues and higher number of features by combining distances and dihedral angles and in both cases, we did not obtain any improvement in the predictions (S11 Fig). In addition to that, we did not observe any particular impact of five set of distances into the predictions as removing them did not decrease the correlations significantly (S11 Fig). On the other hand, knowledge about different amino acids implicitly contains information about amino acid sizes and physical characteristics such as charge and hydrophobicity. Taken together, this may be sufficient to characterize the effects of mutations with respect to phenotypic outcomes.

## VAE models of MD data provide structural classification of mutants

Finally, we developed a VAE model (three layers with 128, 64, and 32 nodes and an attention layer after 32 nodes in both encoder and decoder) based on the MD distance data to identify

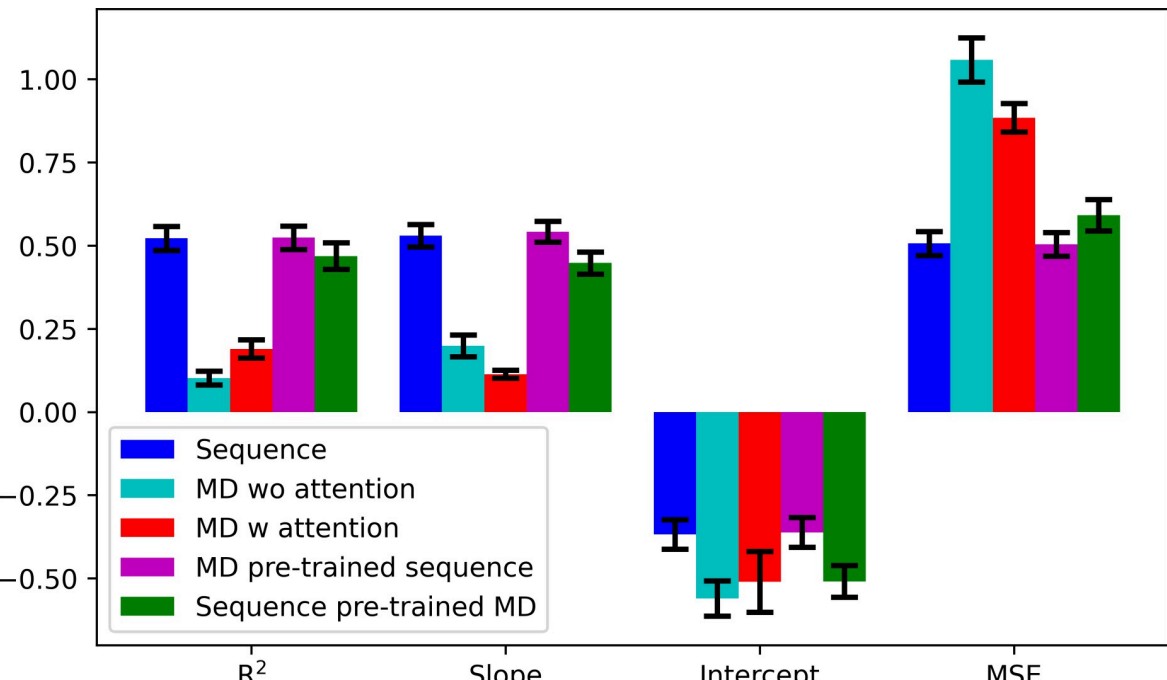

**Fig 5. Performance of phenotype predictions from sequence and MD data based on linear regression $R^2$ correlation coefficients, slope and intercept of linear regression curves, and mean squared errors (MSE) between given and predicted phenotypes.** Every metric was calculated as the average over 10 sets.

structural principles underlying different phenotypical outcomes since the sequence-based ML predictor developed above does not provide such insight. MD simulations of the mutant library with 135 mutants generates a large amount of data that make it challenging to reach generalized conclusions for the structural effects of certain phenotypes. In order to reduce this high-dimensional data, *i.e.* the intra-molecular distance network, into a low-dimensional space we used a VAE model. VAEs are generative models that learn optimal collective variables in the latent space representation, from which higher-dimensional data can be reconstructed with minimal loss via the non-linear decoder block of the VAE model. The projection onto the latent space was then further analyzed via clustering (using Kmeans). Applying the decoder to cluster centers and interpolations between them in latent space then provides information about the key structural determinants that giving rise to different phenotypes. The VAE model was applied as described previously [49], but we also added an attention layer both on the encoder and decoder sides to better account for the variations of effects of mutations on distances as different mutants affects different parts of the enzyme. The resulting latent space mapping is shown in Fig 6A. Mutants in latent space were further grouped into three clusters using a Kmeans clustering algorithm. Although prediction models from MD could not predict phenotypes well, the VAE models did result, to some extent, in a phenotypical separation of mutants as average phenotypes between the clusters I, II and III differed with average phenotypes of -0.49, -0.75 and -0.83, respectively (see S2 Table). Cluster I has a mixture of phenotypes with GOF and LOF mutants as the majority, and cluster II and III contains a majority of lethal and LOF and a minority of GOF mutants.

We proceeded to identify cluster centers in latent space and then applied the generative decoder model to the cluster centers to obtain representative distance information for each cluster. Cluster centers are expected to be meaningful and representative points because of the

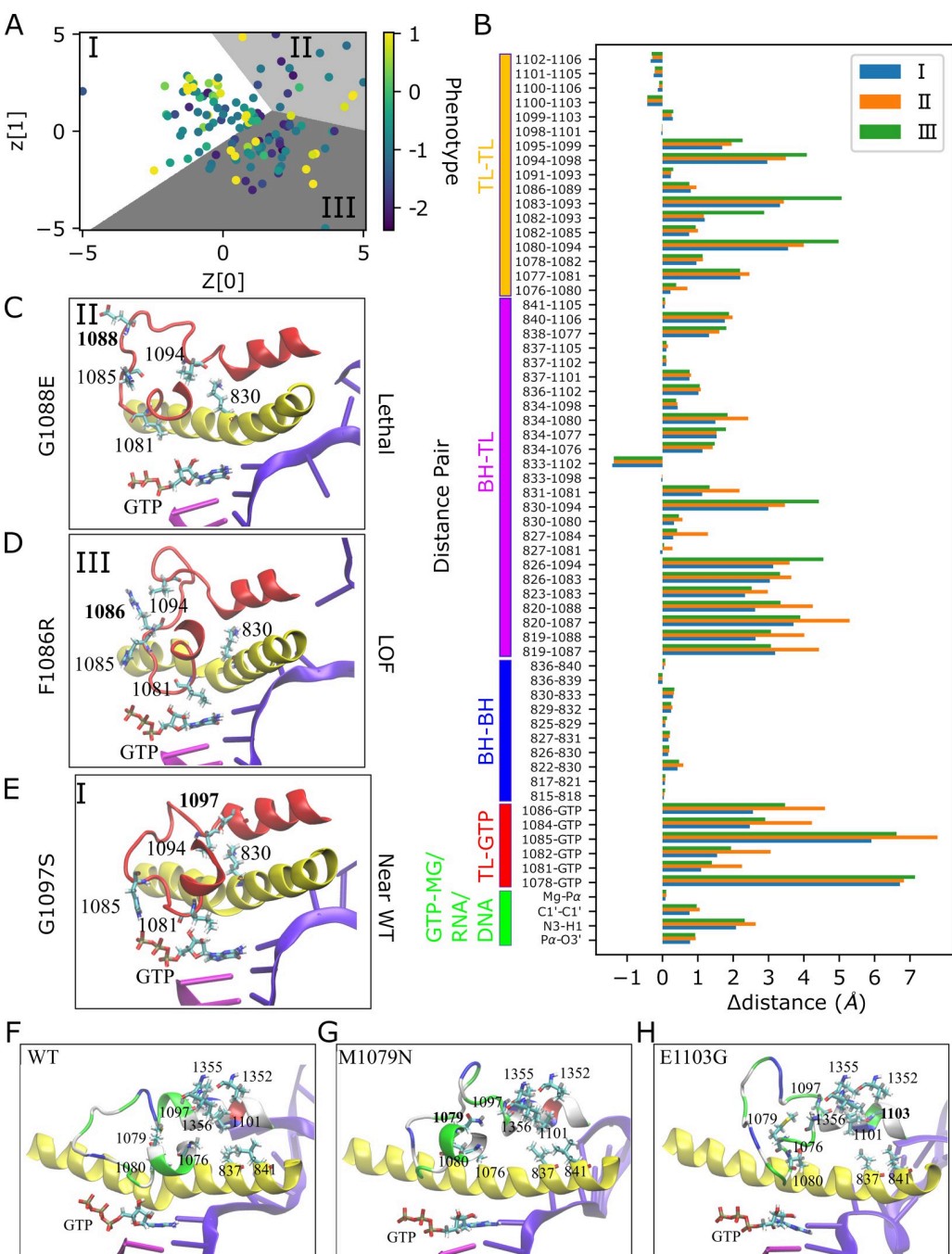

**Fig 6. VAE analysis of MD distance data.** (A) Latent space from VAE model based on MD distance data shaded according to clusters (I is white, II is gray, III is dark gray) and with each point colored to the mutant phenotype. The axes z[0] and z[1] are the first and second dimensions of the 2D latent space generated by projecting the MD data using the VAE model. (B) Distance difference graph; Δdistance shows the differences of the distances from the generative model of VAE at the cluster centers from the WT structure after the equilibration. Panels (C), (D) and (E) show the active site neighborhoods for representative mutants for each cluster in final MD snapshots with the TL (red), BH (yellow), DNA (violet), RNA (magenta) and GTP. Atoms are colored by atom name for residues shown in licorice representation. Panels (F), (G) and (H) show the residues forming hydrophobic pocket for the WT after equilibration and GOF mutants at their final MD snapshots. Color codes are the same as described above except that the TL is shown with residues colored by residue type where positively charged amino acids are in blue, negatively charged ones are in red, polar ones are in green, and hydrophobic ones are in white.

regularization term applied to latent space distribution, which prevents overfitting and helps to obtain meaningful outputs outside the real points. We note that the real points closest to the cluster centers, which are 1076N (phenotype is 1.02), 1080W (phenotype is -2.14) and 1081N (phenotype is -1.85) for the clusters I, II and III, respectively, provided an almost identical distance profiles with the cluster centers suggesting that the latent space is well-regularized. Fig 6B shows the difference map between WT and the resulting distances of each cluster based on the latent space center coordinates. Fig 6B shows that the distances for cluster I, which is dominated by WT-like and GOF mutants, are generally lower than for the other two clusters with more lethal and LOF phenotypes. The finding of longer intra-molecular distances for key residues around the TL for LOF/lethal mutants (Fig 6B) is similar as to what is shown above from the direct analysis of MD data (Fig 4C). However, more detailed insights can be extracted from the VAE-based analysis (Fig 6B): Cluster II features larger distances between TL residues and GTP compared to the other clusters, especially for H1085 and L1081 suggesting that the underlying mutants (T1077N, R; T1080W; F1084D, N; F1086S; G1088E; S1091W; N1106F; all the distances provided in S2 Spreadsheet) act by directly distorting the active site and thereby hindering catalysis. Cluster III also features larger distances between TL and GTP for some mutants (Q1078C, K, S, W; L1081A; H1085L, Q, S, W; A1087Y, V1089G, P1099D; S2 Spreadsheet). However, cluster III mostly features increased distances for BH-TL pairs (Fig 6B), especially for the distances of the residues K830 and D826 with V1094 that are known to be in close distance in the closed TL [4]. This suggests a more indirect mechanism for affecting the mostly negative phenotypes in this cluster that involves disrupting the BH-TL interactions. The BH-TL interactions have been suggested to be functionally important previously [7,32]. In cluster III, most of the substitutions causing large K830-V1094 distances are at close distance to BH, while there are also mutants that are away from BH and results in large distances between K830 and V1094 (S12 Fig). Some of the TL-TL distances for the residues that are close to V1094 are also increased for the cluster III like the distances of T1083-K1093, N1082-K1093, T1080-V1094 (Fig 6B). This suggests, furthermore, a particularly important role of V1094 for the transcription mechanism. Individual members of each clusters show a similar trend with the cluster centers (S13 Fig); the members of cluster II, including the mutants distant to the GTP that are N1106F, S1091W and G1088E, show larger distances for GTP and trigger loop residues while the members of cluster III (F1086R, Q1078K, L1081Y, A1087K and T1080K given in S13 Fig) have higher TL-BH distances for mutants both close to BH (L1081Y, A1087K and T1080K) or away from BH (F1086R and Q1078K).

To further illustrate structural details, we selected three representative mutants for each cluster based on the following key distances: 1085-GTP, 1081-GTP, 830–1094. Accordingly, we selected the G1088E, F1086R and G1097S with predicted numerical phenotypes of -1.75, -0.92 and 0.33 based on fitness data, for clusters II, III, and I, respectively. G1088E would be considered a lethal mutant (phenotype ≤ -1.5). As a result of the mutation, there are large distances between GTP and 1085 (Fig 6C) which are expected to inhibit enzyme function. F1086R is a LOF mutant where distances between GTP and residues 1085 and 1081 remain relatively close, but the distances between BH and TL increase as seen for the pair of 1094 and 830 (Fig 6D). As a result, catalysis is still likely possible but less efficient. Finally, G1097S exhibits a near-WT phenotype as it has 0.33 predicted numerical phenotype, which can be classified as weak GOF-near WT. In this mutant, all residues surrounding the active site are close, resulting in the tighter active site geometry (Fig 6E) that is necessary for WT-like enzyme performance.

GOF mutants are scattered mostly between clusters I and III suggesting that there may also be different mechanisms for this phenotype. The main difference is that the GOF members of cluster III demonstrated relatively larger distances for BH-TL residues compared with GOF

mutants in cluster I (S14 Fig). We selected two mutants, M1079N and E1103G in the clusters III and I, respectively, as representative examples. Their final snapshots are shown in Fig 6F–6H with a focus on the hydrophobic pocket formed by the TL residues A1076, M1079, T1080, G1097 and L1101. Both M1079N and E1103G lead to disruptions of the hydrophobic pocket although the mechanism may be slightly different. M1079 points directly to the hydrophobic pocket with close distances to V1355, I1356 and L1101 in the WT equilibrated structure. Mutation of M1079N directly disrupts the hydrophobic interactions. The last snapshot of the E1103G simulation also shows a disrupted hydrophobic pocket with the M1079 pointing away, but because 1103 is positioned further away, the effect must be more indirect by modulating TL-BH interactions. These results support the hypothesis that the primary mechanism for GOF phenotypes is via the direct or indirect disruption of the hydrophobic pocket formed by TL residues near the BH as suggested by previous studies [30,32]. To quantitatively analyze the hydrophobic pocket, we calculated the number of contacts between the residues (I837, L841, A1076, M1079, G1097, L1101, V1352, V1355, I1356) that form the hydrophobic pocket and the surface area within these residues (S15 Fig). The distributions of number of contacts and surface area were not significantly different among the phenotypes; therefore, it is difficult to reach a generalized conclusion. However, a subset of GOF mutants (L1101E, G1097H, T1080L, L1101H and G1097D) tend to have larger surface area while a subset of LOF and lethal mutants (L1081A, Q1078S, P1099N, V1098R and N1106F) tend to have smaller surface area that may suggest disruption of hydrophobic pocket was observed more in some of the GOF mutants than LOF and lethal mutants.

## Discussion

In this study, we applied ML approaches to interpret genetic fitness values, sequence information, and MD simulation data to predict and characterize TL mutants of yeast RNA Pol II. Fitness values from different conditions were used to generate a quantitative score for TL mutants on a phenotypic continuum from GOF to LOF and lethal. Then, we asked if machine learning approaches using protein sequence and MD simulation could predict these phenotypes when trained on a subset of mutants. The amino acid sequence of proteins is widely used with machine learning approaches to predict various information about structure [50,51,52,53,54] and function [55,56,57] to mutational phenotypes [35,37,40,42]. These studies suggest that sequences contain the key information about the structure and function of the proteins, which can be learned. In this study, we used the sequence of TL residues as an input and obtained predictive models of the mutant phenotypes that provide higher correlations than a simple model from average phenotypes for each residue. It may be possible to improve these models further by incorporating additional data on basic physical properties of mutation sites like molecular weight and volume, hydrophobicity, surface area, solvation energy, electrostatic interactions, position specific scoring matrix (PSSM), *etc.* as applied in some of earlier studies [35,37,58]. Similar approaches could be used for RNA polymerase or other systems to understand function and phenotypes, especially for disease-related mutants. We also tested the sequence-based models trained on single mutants for the prediction of double mutant phenotypes. The model predicts the addition of similar type phenotypes and suppression of opposite type phenotypes and it predicted lethal phenotypes for some LOF-LOF double mutants like H1085Q-Q1078S, but was unable to predict lethal phenotypes from the combination of GOF-GOF mutants observed previously [21]. Prediction could easily be improved by training with double mutants allowing the model to learn different double mutant effects.

Recent studies showed that the combination of MD simulations with ML algorithms could provide insight on dynamics, conformations, and kinetics of proteins [59,60,61,62,63]. With

the motivation from these studies, we used the distance data from MD simulations to investigate their predictive performance. There are computational limitations of working with a large protein like RNA Pol II. Thus, instead of running long simulations, we performed multiple, relatively short simulations to obtain insight about the structural effects of mutations. Surprisingly, we found that structural data obtained from these MD trajectories could not add predictive abilities with respect to the functional phenotypes for specific mutations when used within a ML framework. In light of this finding, several potential advances can be imagined. First, longer, perhaps on μs scales, or higher-quality simulations with different force fields might allow greater inference on mutant mechanisms. Second, obtaining additional data from simulations using a starting structure with an open TL in addition to the closed TL structure and simulating the transition from open to closed states of TL may be needed to provide more detailed structural, physics-based input in addition to what is already encoded in differences in amino acid sequences when predicting enzyme function. Simulation of the transition from open to closed TL are computationally expensive especially for a large mutant library, while such simulations may provide a deeper understanding of the functions of mutants and will be a future direction of this study. Our previously published study on open and closed TL exhibited that the distances for TL and BH residues are dynamically changing along the transition from open to closed state [12]. Some of the TL-TL and TL-BH distances are larger in the open TL compared to the closed TL, while the others showed smaller distances for the open TL or the transition states [12]. Additional analysis on open and closed state simulations based on our published study [17] showed that most of the distances we used in the machine learning analysis, especially for TL-GTP and TL-BH pairs, were larger for the open TL (S16 Fig) indicating that those distances are relevant to the closed TL. Notably, the distances that are larger in the open TL also appeared to be increased for the lethal and LOF mutants (Fig 6B) suggesting that lethal and LOF mutants were destabilizing the closed TL. The previous studies also showed that some of the TL residues had overall large distances for the closed TL compared to open TL. We speculate that such TL residues, like E1103 or 1093, may have impacts on the open TL state and the phenotypes of the mutations of these residues may have structural outcomes for the open TL simulations rather than the closed TL. Therefore, the mutant simulations of such residues on the closed TL state may not provide a relevant information about their function and this might be partly responsible for the noisy results of the ML models.

Despite the limitation in the MD data, VAE models developed based on MD still could provide mechanistic insights into the potential structural basis of lethal, LOF, and GOF mutants. Based on this analysis, it appears that a subset of lethal/LOF phenotypes (cluster II in Fig 6A and 6B) correlates with mutations that directly increase the distances between key TL residues and the GTP in the active site, thereby inhibiting catalysis. Another subset of lethal/LOF phenotypes (cluster III in Fig 6A and 6B) for the examined mutants appears to be related more to disruptions in the TL-BH interactions. In contrast, some of the GOF phenotypes appear to result from a disruption of a hydrophobic pocket near the TL (Figs 6F–6H and S15). With a compromised hydrophobic pocket, closing of the TL may be favored, accelerating enzyme kinetics presumably at the expense of reduced fidelity. We expect that mutations on residues outside the TL and in close distance to the hydrophobic pocket like V1352, V1355, I1356, which were not tested in any earlier studies, may manifest GOF phenotypes and provide additional insight of the function of this hydrophobic pocket in achieving GOF phenotypes. Additionally, we observed increased BH-TL distances (Cluster III in S14 Fig) in a subset of GOF mutants. These general findings are consistent with previous studies but go further because of a more comprehensive analysis that is based on a systematic analysis of a larger number of TL mutations. Still, more work is left to be done to understand specific mutations and specific roles of individual residues. One avenue for further studies may be via the exploration of

double mutants that may restore non-WT phenotypes to WT function based on the predictions made above.

## Conclusion

In this study, we report a comprehensive characterization of the mutations of RNA Pol II TL residues using ML techniques on high throughput genetic fitness data, sequence data and MD simulations of TL residue mutations. Our study suggests that fitness data and sequence information are correlated such that the phenotype that was predicted from fitness values can be learned by sequence information and the predictions from sequence go beyond the simple prediction model from average phenotypes. Such a prediction was not possible with our MD data due to computational limitations. Nevertheless, MD data could provide some mechanistic understanding on different phenotypes. Longer simulations may be necessary to obtain a predictive model that would be comparable with the sequence-based model. However, μs scale simulations of large TL mutation libraries of RNA Pol II still remain a major computational challenge. As an alternative to that, artificial intelligence methods that predict structural details from sequence of mutants could be developed and applied to RNA Pol II systems as a future direction.

## Methods

### Prediction of continuous phenotypes from fitness data

The Pol II TL fitness and phenotypic landscape approach of Qiu *et al.* [32] based on a deep mutational scanning approach was applied here to a second-generation TL mutant library [64]. Briefly, a library of mutants is grown under different experimental conditions. These conditions detect growth changes of mutants relative to WT (phenotypes) that are predictive of Pol II biochemical defects [32]. Phenotypes are expressed as "fitness" scores that represent the log2 allele frequency changes of individual mutants within the library pool over time for the series of experimental stress conditions relative to a control growth condition relative to allele frequency changes of a WT allele. Allele frequencies are determined by deep sequencing of variant pools after growth relative to starting or control conditions. Sequencing allows quantitative determination of each allele's frequency in the library. A mutant that grows worse than WT for a particular stress condition relative to control will have a negative fitness score whereas a mutant that grows better than WT for a particular stress condition will have a positive fitness score.

The construction of the TL mutant library was followed by *en masse* phenotyping assays monitored by deep sequencing. This work will be described in detail elsewhere but key updates to the original approach are as follows. A second-generation TL mutant library was synthesized using programmed oligo synthesis (Agilent). Library oligos were amplified from synthesized pools and homology arms of WT sequence were added using overlap PCR. These fragments comprising ~200 nt of flanking *RPB1* sequence on each side and a central 93-nt region encoding the WT Pol II TL (Rpb1 amino acids 1076–1106) or individual TL variants were introduced into yeast along with *RPB1* plasmid lacking TL sequence and linearized at the TL position in three replicates. This cotransformation allows construction of a pool of variant plasmids by gap repair, exactly as performed previously by Qiu et al. Transformants were plated at high density (~10,000 per plate) instead of 300–400 as done previously. 5-FOA-resistant colonies were scraped from SC-Leu+5FOA plate and replated on SC-Leu, SC-Leu + 20mg/ml MPA (Fisher Scientific), SC-Leu + 15 mM Mn (Sigma), YPRaff, YPRaffGal, SC-Lys, and SC-Leu + 3% Formamide (JT Baker) plates for phenotyping. Cells were scraped from each phenotyping plate after defined growth periods and genomic DNA was extracted from each screening plate with Yeastar Genomic DNA kit (Zymo) and amplified using

emulsion PCR (EURx Micellula DNA Emulsion & Purification (ePCR) PCR kit) according to manufacturer's instructions. A dual indexing strategy where custom indexing primers were paired with primers using 28 NEB indices was utilized in the amplification to discriminate between various screening plates. Amplified libraries were sequenced with Illumina Next-seq for 150nt single-end reads.

Experimental fitness data for yeast Pol II TL variants from a total of 21 conditions with three replicates each were used for deriving a predictive phenotype model. Missing fitness values were imputed using mean values of a given feature. The model was based on continuous real-valued phenotypes, where previously classified GOF, LOF, and lethal outcomes map to values of +1.0, -1.0, and -2.0 and where the WT maps to 0.0. A neural network consisting of three fully-connected layers with 256, 128 and 64 nodes (S17A Fig) was trained against known phenotypes from previous studies [3,21,32] for 83 mutants. Mutations that did not lead to clear GOF, LOF, or lethal outcomes were treated as WT with a continuous phenotype value of 0.0. The mean squared error (MSE) was used as the loss function. After training based on the classified mutations, phenotypes were predicted from the experimental fitness data for all of the TL mutants.

The fitness data was used further in a variational autoencoder model to map the phenotypes into a reduced dimensional space. Both encoder and decoder models have three layers with 256, 128 and 64 nodes (S17B Fig). The loss function of MSE between input features and generated output values was used. For the regularization of the latent space, the Kullback-Leibler (KL) divergence between latent space distribution and the standard normal distribution was applied as defined by Kingma and Welling [49]. The performance of the generative models with 2D and 3D latent spaces is similar (S1 Fig). All the fitness data, the predicted phenotypes, and the latent space coordinates are summarized in the S1 Spreadsheet.

## MD simulations

The WT RNA Pol II structure used as a starting point was deposited to Protein Data Bank with PDB ID:2E2H [4]. Missing residues were modeled for the loops that have less than eight amino acids using MODELLER version 9.15 [65]. The histidine at 1085 of Rbp1 was protonated based on the study by Huang et al [18]. The system was solvated in a cubic box with a 10 Å cutoff distance between the box edges and any atom of the RNA-Pol II complex resulting in a total box size of 162 Å. The system was then neutralized with $Na^+$ ions. Periodic boundary conditions were applied along with the particle-mesh Ewald Algorithm for the calculation of long-range electrostatic interactions. Lennard-Jones interactions were switched from 10 to 12 Å. The SHAKE algorithm was used to constrain bond lengths involving hydrogen atoms. The CHARMM 36m force field [66] was used for proteins and the CHARMM 36 force field [67] was used for nucleic acids. The TIP3P model [68] was used for explicit water molecules and a recently suggested NBFIX for $Na^+$ phosphate interactions was applied [69]. The force fields were modified to redistribute the atomic masses of atoms that are attached to hydrogen atoms, so that hydrogen atoms had an increased mass of 3 a.m.u instead of 1 [70]. This modification allowed us to perform the simulations using a 4 fs time step.

The WT system was subjected to 5,000 steps of energy minimization. The system was then equilibrated for around 1.6 ns by gradually increasing the temperature from 100 K to 300 K and using restraints on the heavy atoms of backbone and sidechains with force constants of 400 and 40 kJ/mol/nm$^2$, respectively. The equilibrated Pol II complex was used to prepare the single site TL mutants. In total, 135 mutants were prepared (see S2 Spreadsheet). Each mutant system was minimized again using the same minimization scheme as for the WT and equilibrated for an additional 1 ns. Production simulations were then performed using Langevin

 

dynamics with a friction coefficient of 0.01 ps$^{-1}$ under the constant temperature of 298 K. Simulations were run using OPENMM [71] on GPU hardware. 100 ns production runs were performed for three replicates for each mutant and the last 50 ns of the production runs was used for the analysis. In total, 40.5 μs of mutant simulations were generated.

The analysis of the MD simulations was done using the MMTSB package [72] in combination with in-house scripts. The minimum distances between the residues were analyzed and the average distances were calculated from the combined distribution of distances from the three replicate simulations. The distances for the hydrophobic pocket were calculated as minimum pairwise distances between the residues I837, L841, A1076, M10079, G1097, L1101, V1352, V1355, I1356. Two residues were considered to be in contact if they were within a distance of 6 Å. All the distances and numbers of contacts are provided in Supplementary Sheet 2. Solvent accessible surface area for the hydrophobic pocket was calculated using MMTSB package with probe radius of 1.4 Å. The RMSD was calculated for the TL and BH regions for all the mutants and is shown in S18 and S19 Figs, respectively.

## Phenotype prediction models using sequence and MD data

For models predicting phenotypes based on the TL amino acid sequence, sequence data was converted into one-hot encoding sparse matrices, where each feature was a 21-size vector that represents a single amino acid along the sequence. Histidine was coded as either protonated or deprotonated to allow the model to distinguish protonated histidine at residue 1085. The one-hot-encoded sequence-based features, a two-dimensional matrix (31x21), were used as input for a neural network with three fully connected layers with 128, 64, and 32 nodes in two dimensional matrices (31x128, 31x64, 31x32), respectively. The 32-node third layer was flattened two a one-dimensional vector with a size of 992 and passed through another layer with 32 nodes before the single-valued output layer (S17A Fig).

For models trained to predict phenotypes based on MD data, pairwise distances between key residues involving the TL and neighboring elements were extracted from the simulation trajectories and averaged. A list of distances used in the models is provided in the S2 Spreadsheet. A neural network with three fully connected layers with 128, 64 and 32 nodes was used as for the sequence-based predictors. To emphasize the effect of mutations at the different parts of the structure, we also tested another model that has 128 and 64-nodes layers that were connected to the output layer via an attention layer [73] (S17A Fig). Attention is a more complex machine learning model that can learn to focus on the most informative part of the data and fade out other parts that are less important as a function of the input data. The attention layer was generated for the last hidden layer with 64 nodes, which was used as query, key, and value vectors for the self-attention framework. The new values were calculated by the multiplication of the values and the weighted sum of the similarities between the query and key vectors.

The mutants examined via MD simulations were used for both sequence and MD prediction models to compare the performances of the neural network models. The mutants were split into ten training and test sets with randomly chosen 100 and 35 mutants, respectively. A 1:1 combination of loss functions of mean squared error (MSE) and KL divergence was applied to the prediction models with sequence and MD data. The KL divergence was calculated analytically as follows:

$$KL = \frac{1}{2}\left(\left(\frac{\sigma_{True}}{\sigma_{Pred}}\right)^2 + \left(\frac{(\mu_{True} - \mu_{Pred})^2}{\sigma_{Pred}^2}\right) - 1 + \log\left(\frac{\sigma_{Pred}^2}{\sigma_{True}^2}\right)\right) \tag{1}$$

The model weights and biases were saved every 500 epochs and training continued until a maximum of 20,000 epochs. For each set, ten different models were generated, and the best model was chosen from the saved models at intervals that provides the best combination of $R^2$ correlations and slope for the regression line between predictions and label phenotypes for the test set. The overall performance of the models was calculated by the average performance of the best models of the sets for the comparison of different models from either sequence or MD data. A complete phenotypical landscape was generated using the sequence-based models based on the ensemble average of the predictions from the best models of the sets.

The models based on MD data and sequence were also combined to see if there was any improvement in the predictions when using both data as input features. The data was combined in two different ways: First, we took the pre-trained sequence model and added the MD-based model to be trained to add additional input to the last layer of the sequence model before generating the output. Second, we used a pre-trained MD model and added the sequence model and concatenated the last layers of the two models to generate the final output. In each case, we froze the pre-trained weights to test whether the additional input features can improve the predictions.

## Variational auto-encoder models based on the MD data

Variational auto-encoder models were generated based on the MD data in order to extract mechanistic insights from the simulations. Both, the encoder and decoder networks, consisted of three fully-connected layers with 128, 64, and 32 nodes. We applied an asymmetrical VAE model at which the attention layers were applied after the layers with 32 nodes for both, the encoder and decoder (S17B Fig). Attention layers help the model to better group similar phenotypes together compared to the models without attention (S20 Fig). We experimented with 2D and 3D latent spaces. The performance of the generative model with a 3D latent space was similar to the model with the 2D latent space (S21 Fig). Therefore, we performed further analysis of the model with the 2D latent space. The resulting reduced dimensional latent space was then clustered via a Kmeans clustering algorithm as it provided a better visual separation than other clustering algorithms (S22 Fig). Three clusters provided the best separation of average phenotypes between the clusters (S23 Fig). Thus, we separated the latent space into three clusters. The cluster centroids were used subsequently to generate representative molecular states for each cluster via the decoder network.

## Machine learning details and software

All ML models were generated with the Tensorflow package [74]. The models are summarized as diagrams in S17 Fig. The Python scripts to train the models and to predict from the trained models along with the weight of the trained models and the input files for the models are available at https://github.com/bercemd/PolII-mutants. Data imputation, Kmeans clustering and principal component analysis (PCA) were performed with the Sklearn module in Python [75]. The Adam optimizer was used for all models. The learning rate and number of epochs were varied according to the model that are summarized in S1 Table. Different learning rates were applied to prevent unstable training but achieve convergence to a minimum loss within reasonable training times (S24–S26 Figs). A batch size of 4 was used for all models except for the prediction models with the KL divergence loss in which a batch size of 100 is used to calculate the loss for the complete training set. The rectified linear unit (ReLU) activation function was used for each hidden layer for both prediction and VAE models except for the attention layer, where the Softmax activation function was used.

## Supporting information

**S1 Fig.** The distribution of mutants on the latent spaces (top) and generative performances (bottom) of the VAE models with 3D (left) and 2D (right) latent spaces using fitness data as the input.
(TIF)

**S2 Fig. Principal component analysis (PCA) on the fitness data with each data point colored according to its corresponding phenotype.**
(TIF)

**S3 Fig. Latent space of the unsupervised VAE model based on the fitness data.** Each data point is colored according to its corresponding phenotype with transparency except the selected mutants that are colored without transparency for clarity. GOF mutants and LOF/lethal mutants at the boundary are shown. The position of E1103G in the latent space is also shown since its double mutants with the GOF mutants at the boundary cause lethality.
(TIF)

**S4 Fig. Training and test loss during the training of the models.** The models are trained with the input features from sequence data (Sequence), MD data without an attention layer (MD wo attention), MD data with an attention layer (MD w attention), MD and sequence data with pre-trained sequence weights (MD pre-trained sequence) and pre-trained MD weights (Sequence pre-trained MD).
(TIF)

**S5 Fig. Correlations between predictions and label phenotypes for the training sets of different models.** The model details are as in S2 Fig.
(TIF)

**S6 Fig. Correlations between predictions and label phenotypes for the test sets of different models.** The model details are as in S2 Fig.
(TIF)

**S7 Fig. The difference map for the phenotypes predicted from sequence and fitness.** Y-axis shows the absolute value of the differences of phenotypes from the fitness and sequence data and X-axis shows the phenotypes from the fitness data. The outliers with difference larger than 1.25 are shown.
(TIF)

**S8 Fig. Prediction of phenotypes from the average phenotypes for each mutant for the training sets.** (A) the phenotypes predicted from the average values vs phenotypes from the fitness and the linear regression line (B) the average phenotypes from the training sets shown in the complete mutation map.
(TIF)

**S9 Fig. Analysis of phenotype predictions of double mutants using sequence-based ML models.** (A) Predicted phenotypes ($P_{predicted}$) vs. additive phenotypes ($P_{additive}$) that were calculated as the sum of the predicted phenotypes of single mutants, (B) the change of difference between $P_{predicted}$ and $P_{additive}$ with respect to the spatial distance between the mutation sites. Each point is colored with the single phenotype before the second mutation site (E1103G, G1097D, F1084I or Q1078S) was introduced.
(TIF)

**S10 Fig.** Heatmap plots of phenotypes vs average distances for the five groups of distances that are the distances between 1) GTP and MG/RNA/DNA, 2) GTP and TL residues, 3) BH and BH residues, 4) BH and TL residues, and 5) TL and TL residues for the mutants from MD simulations. (TIF)

**S11 Fig. Performance of phenotype predictions from MD data based on linear regression $R^2$ correlation coefficients.** Eight model was generated using different inputs from MD data that are the five set of distances near the active site (see Fig 4), TL backbone dihedral angles, combination of distances and dihedral angles and distances by excluding one set of distance values at each model. (TIF)

**S12 Fig. The average distances between K830 and V1094 in mutant simulations vs. the minimum distance of the mutated amino acid site to the BH in the WT structure.** Each panel shows the plot for the members of each cluster found in the MD-VAE latent space. The dashed lines show the distance of K830 and V1094 in the WT structure. (TIF)

**S13 Fig. Distribution of distances between L1081 and GTP, H1085 and GTP, and K830 and V1094 for the selected members of the clusters from the VAE latent space.** The mutants are given in the legends with their continuum phenotypes in the parentheses. (TIF)

**S14 Fig. Distribution of distances between D826 and V1094, K830 and V1094, and G819 and G1088 for the selected GOF mutants of the clusters from the VAE latent space.** The mutants are given in the legends with their continuum phenotypes in the parentheses. (TIF)

**S15 Fig. Analysis of the hydrophobic pocket formed by the residues I837, L841, A1076, M1079, G1097, L1101, V1352, V1355, I1356.** (A) Number of contacts vs. phenotypes at the hydrophobic pocket. (B) Surface area vs. phenotypes within the hydrophobic pocket. (TIF)

**S16 Fig. Distances from the open and closed TL simulations of WT Pol II.** Distances were calculated by analyzing the simulations published in an earlier study. (TIF)

**S17 Fig. The diagram of the neural network models.** (A) The models for the prediction of continuous phenotypes have alternating layers depending on the input: For the models with fitness score as the input, three dense layers were used. For the models with the MD data as the input, three dense layers or two dense layers and one attention layer were used. For the models with amino acid sequence as the input, two-dimensional matrix at the third dense layer was flattened out and passed through another dense layer. (B) VAE model was applied to the fitness scores as three dense layers on the encoder and decoder models. It was applied to the MD data with additional attention layers on the encoder and decoder. (TIF)

**S18 Fig. RMSD values of TL residues for 135 mutants.** RMSD values from three replicate simulations were represented with different colors. There are not large changes in RMSD for TL suggesting that TL is retaining its overall conformation for the mutants within the simulation time scale. (TIF)

**S19 Fig. RMSD values of BH residues for 135 mutants.** RMSD values from three replicate simulations were represented with different colors. There are not large changes in RMSD for BH suggesting that BH is retaining its overall conformation for the mutants within the simulation time scale.
(TIF)

**S20 Fig. The distribution of mutants on the latent spaces generated by VAE with and without attention layer using MD data as the input.**
(TIF)

**S21 Fig.** The distribution of mutants on the latent spaces (top) and generative performances (bottom) of the VAE models with 3D (left) and 2D (right) latent spaces using MD data as the input.
(TIF)

**S22 Fig. The clustering of the VAE latent space of the MD data using different clustering algorithms.** Each cluster is shown in colors from white to different shades of grey; mutants are scattered, and color coded with corresponding phenotypes.
(TIF)

**S23 Fig. The clustering of the VAE latent space of the MD data using Kmeans clustering algorithm with three, four and five clusters.** Each cluster is shown in colors from white to different shades of grey; mutants are scattered, and color coded with corresponding phenotypes; at each cluster the average phenotypes of the mutants are shown.
(TIF)

**S24 Fig. Three replicates of VAE models using the fitness data as features at different learning rates.** Models with $10^{-6}$ learning rate were not converged. Models with $10^{-3}$ learning rate tend to be stuck in a local minimum loss. The models with $10^{-4}$ and $10^{-5}$ learning rates provided similar latent spaces without any convergence problem, therefore a learning rate of $10^{-4}$ was used for fitness-based models.
(TIF)

**S25 Fig. Three replicates of VAE models using the MD data as features at different learning rates.** The same trend with the S20 Fig is observed. Learning rate of $10^{-4}$ provided the most visual separation of phenotypes, therefore it was used for the MD data models.
(TIF)

**S26 Fig. Three replicates of prediction models using the sequence data as features at different learning rates.** Learning rate of $10^{-5}$ provided the minimum losses for the test sets, therefore it was used for the sequence models.
(TIF)

**S1 Table. Summary of deep learning models.**
(DOCX)

**S2 Table. Statistical analysis of the clusters obtained from VAE model using MD data.** T-test was performed by assuming the two populations have different variance. Clusters I, II and III corresponds to the clusters shown in Fig 6A.
(DOCX)

**S1 Spreadsheet. The fitness scores, the predicted phenotypes, and the latent space coordinates.**
(XLSX)

**S2 Spreadsheet. The distance analysis results, and the latent space coordinates from the distance input features from MD simulations.**
(XLSX)

## Acknowledgments

We thank Guillermo Calero for helpful discussions. We used computational resources at the Institute for Cyber-Enabled Research/High Performance Computing Cluster (ICER/HPCC) at Michigan State University and at the National Science Foundation's Extreme Science and Engineering Discovery Environment (XSEDE) facilities.

## Author Contributions

**Conceptualization:** Bercem Dutagaci, Craig D. Kaplan, Michael Feig.

**Data curation:** Bercem Dutagaci, Bingbing Duan, Chenxi Qiu, Craig D. Kaplan, Michael Feig.

**Formal analysis:** Bercem Dutagaci, Bingbing Duan, Chenxi Qiu, Craig D. Kaplan, Michael Feig.

**Funding acquisition:** Craig D. Kaplan, Michael Feig.

**Investigation:** Bercem Dutagaci, Bingbing Duan, Chenxi Qiu, Craig D. Kaplan, Michael Feig.

**Methodology:** Bercem Dutagaci, Bingbing Duan, Chenxi Qiu, Craig D. Kaplan, Michael Feig.

**Project administration:** Bercem Dutagaci, Craig D. Kaplan, Michael Feig.

**Resources:** Craig D. Kaplan, Michael Feig.

**Software:** Bercem Dutagaci, Michael Feig.

**Supervision:** Craig D. Kaplan, Michael Feig.

**Validation:** Bercem Dutagaci, Bingbing Duan, Chenxi Qiu, Craig D. Kaplan, Michael Feig.

**Visualization:** Bercem Dutagaci, Michael Feig.

**Writing – original draft:** Bercem Dutagaci, Craig D. Kaplan, Michael Feig.

**Writing – review & editing:** Bercem Dutagaci, Bingbing Duan, Craig D. Kaplan, Michael Feig.

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
