## [Decision Letter · Decision Letter 0]

27 Sep 2022

Dear Dr. Feig,

Thank you very much for submitting your manuscript "Characterization of RNA Polymerase II Trigger Loop Mutations using Molecular Dynamics Simulations and Machine Learning" for consideration at PLOS Computational Biology.

As with all papers reviewed by the journal, your manuscript was reviewed by members of the editorial board and by several independent reviewers. In light of the reviews (below this email), we would like to invite the resubmission of a significantly-revised version that takes into account the reviewers' comments.

We cannot make any decision about publication until we have seen the revised manuscript and your response to the reviewers' comments. Your revised manuscript is also likely to be sent to reviewers for further evaluation.

Sincerely,

Guanghong Wei

Academic Editor

PLOS Computational Biology

Nir Ben-Tal

Section Editor

PLOS Computational Biology

Reviewer's Responses to Questions

**Comments to the Authors:**

Reviewer #1: In this work, the authors have combined MD simulations and ML techniques to characterize the relation between the trigger loop mutants and the phenotypes. They have found out that the TL sequences could serve as the appropriate variables to predict the phenotypes while inclusion of the structural information extracted from the MD simulations would not further improve the prediction. Moreover, they have applied the variational auto-encoder model to correlate the GOF, LOF or lethal mutations with the key distances, such as the TL-BH distance, the TL-GTP distance, etc, which could provide the structural insights into the influence of the mutations on the phenotypes. Overall, this work is interesting and has applied the ML techniques to the critical enzyme Pol II to make prediction about the TL mutations, which could advance our knowledge about the role of TL in transcription. I think it fits the scope of Plos Comput. Bio.. However, I have several comments for the authors to address.

Major comments:

1. In the introduction, please review how the machine learning (ML) frameworks have been applied to predict the phenotypes upon mutations in other biological systems or justify why you decide to choose ML in the current work.

2. Page 7, line 142: It is not clear if the training set has included the mutations on all the residues on the TL. This information should be provided and the complete training set (including the mutation and the phenotype) could be added in the SI.

3. Page 8, line 147: “… the fact that these TL residues are critical for function.” Please specify how they are critical for what function.

4. Page 8, line150-157: In this part, the authors have discussed the mutations on several amino acids, such as A1076, M1079, G1097, L1101, K1092 and K1093, would destabilize the open TL state and result in GOF, as it would facilitate the closure of the TL. Can the authors perform MD simulations for some mutations to validate their proposal?

5. Page 9: how to define the “transition”? Can they provide some examples for the mutations that lead to the direct transition from lethal to GOF?

6. Fig. 1c: what the z[0] and z[1] stand for?

7. Page 13, line 225: what's the biological outcome when GOF pheonotypes crosses the threshold for viability?

8. Page 14-15, line 253-257: It is difficult to understand their conclusions from Fig. S7. It is not clear why lethal mutants (phenotype < -1.5) have effects mostly on the TL-GTP distances while the distances between TL and BH residues are affected by mostly LOF mutants. Can the authors explicitly explain it?

9. Page 17, line 290-296: as the clustering results in a mixture of phenotypes, does that implicate that the clustering algorithm is not appropriate? How about K-center or K-medoids? My concern is that if the phenotypes are mixed within the cluster, the later analysis about the cluster center may not represent the features of the specific phenotype.

10. Also, for Kmeans, the center may not lie in a real data point and how to calculate the distance for the center in this scenario?

11. A summary or discussion about how to achieve a GOF phenotype by TL mutations would be helpful for the further experimental manipulation.

Minor comments:

1. In the abstract, please spell out “GOF”.

2. Fig. 6B: please color the y-axis in panel B by the type of interaction pair

Reviewer #2: Current ms contains studies using machine learning (ML) methods to analyze RNA polymerase (RNAP) mutants to predict mutant phenotypes from protein sequence and to probe whether molecular dynamics (MD) simulation data can improve the prediction. The RNAP mutants were generated systematically on a key structural element, the trigger loop (TL) of the RNAP, and the authors were able to demonstrate that the TL amino acid sequences can be trained to show R2~0.68 correlation with the phenotype quantities, i.e., an indication of some extent of predictability. In contrast, while the MD simulation (over tens of microseconds in aggregation) data were introduced, the current ML implementations could not really improve the phenotype prediction over that from the protein TL sequences. The work presents well intentions on bringing structural dynamics from all-atom MD to improve the ML performances, and then shows that current implementations have not been able to achieve the intended improvements yet.

It is healthy attitude to show with substantial efforts on the implementations, the performances are not necessarily as expected. On the other hand, I doubt whether the MD simulation datasets produced so far have been exhaustively analyzed to draw t conclusions or provide sufficient controls to help future improvements. Certain insights on the TL mutations to phenotypes or RNAP functions are provided. However, it was not clear whether some of insights were drawn from current datasets (upon the ML implementations), or more or less based on the authors’ prior expertise or knowledges on the RNAP system. In presenting, some of statements made in the text are not well connected with data or no figures were shown to support. Some early part of ML implementations in the absence of the MD simulation data were not sufficiently explained to facilitate learning. Detailed points are provided below for the authors’ consideration of improvements before this ms can be further considered.

1) Introduction (1st paragraph): “Proposed mechanisms for the NAC emphasize conformational changes of a highly conserved domains in the active site, present within the largest subunit of yeast Pol II, Rpb1. One of these domains is called the trigger loop (TL) and the other is the nearby bridge helix (BH).”

Have the authors considered regions beyond TL and BH within the ‘highly conserved domain or domains”, i.e., to be probed for analyses, as current TL and BH part of analyses might not contain sufficient information or notable improvements for the ML implementations?

2) Continued with the above text: “The TL has open and closed conformations, which are known to be important for the nucleotide addition … Upon initial binding of the NTP, the TL closes and catalysis is promoted for substrates base-paired with the template. This results in the pre-translocation (substrate added) state, followed by TL opening together with the pyrophosphate ion (PPi) release, and subsequent or concurrent translocation”.

In the MD simulations conducted, were the TL modeled in the open or closed conformation, or both? It appears the both TL open and closed conformations are important in the nucleotide addition cycle, so catching both the TL open and closed conformations appear to be necessary.

(3) Introduction (2nd paragraph): “In previous studies, a complete deletion of the TL from different species caused marked reductions in transcription rate (23, 28, 29).”

When the TL was completely deleted, were there still marginal transcription activities documented? If so, such activities in the absence of TL would be quite interesting and deserve to be included in the modeling for phenotypes.

(4) Continued with the above text: “On the other hand, E1103 mutations are known to cause an increased catalytic rate but with compromised fidelity”

How did the authors account for the E1103 mutations, only as ‘gain of function’ or GOF, or ‘loss of function’ or LOF, or both? Did such choices of GOF/LOF on the E1103 type of mutations (how many of such types beyond E1103) impact on the ML implementations?

(5) Introduction (3rd paragraph on page 2) “To address this, we combined the data from experimental fitness scores and molecular dynamics (MD) simulations for TLs with different amino acid sequences to predict functional and structural outcomes of TL mutations using machine learning (ML) frameworks.”

How were the experimentally fitness scores measured? Technically, through the ms and SI, I cannot find explanations on such basics. It is important to understand the datasets used in the analyses, so it is necessary to explain all related data measurements. To be more specific, what about the fitness scores measured for those notable mutants mentioned in the Introduction, as one may wonder whether those fitness scores were reasonably constructed in the first place.

(6) Results (1st paragraph) “We describe here three sets of results: 1) Based on fitness data from second generation deep mutational scanning of the Pol II TL in S. cerevisiae, we generated a model for the classification of TL mutants along a continuous phenotypic spectrum; 2) we applied ML to infer TL mutant phenotypes from TL sequence with and without structural data from MD simulations using a subset of gold standard mutants as training data; and, 3) we extracted mechanistic principles for how different classes of TL mutations modulate Pol II function based on variational auto-encoder (VAE) models that were trained on MD simulation data.” and Fig 1.

Further clarifications on the above approaches are needed. The above outlines list VAE models in the third approach. In Fig 1 to present for the first approach, VAE model was also used in Fig 1C. Are VAE techniques instead of some type of ‘models’? In Fig 1C, what is the ‘latent space’ or what are the z[0] and z[1] data? The authors did not explain the jargons “latent space’ or VAE etc., which would bring difficulties to readers who are not ML experts but still want to learn the implementations here. Besides, the fitness data in Fig 1A are shown for different conditions (x-axis) and TL residue number (y-axis), what are those different conditions, and are the fitness data measured for each TL residue but not just for each mutant species? Again, it’s confusing if one wants to understand the fitness data.

(7) Continued with the above text (lines 132-137 on page 7).

In the previous deep mutational scanning, how much percentiles of mutants were classified as GOL/LOF/lethal, respectively? What are the necessities or advantages of using “a continuous phenotypic spectrum” instead of doing classifications (or discretized) as GOF/LOF/lethal etc.? What is the purpose of “project the phenotypes into a two-dimensional latent space and analyze”, just dimension reduction? Why only two but not say three dimension?

(8) Results “Fig. 2A shows the average prediction performance with good correlation (R2 = 0.68)”, would the performance or correlation values change by changing parameters in ML implementations, e.g. how many mutants used for training and how many for testing.

(9) Results (page 12) “Generally, our model predicted additive effects double mutants on phenotypes in which similar phenotypes showed an increase effect while opposite phenotypes suppressed each other”.

Are those residues in mutations showing additive effects spatially separated (or distant) so that to be able to “additive”?

(10) MD simulation studies on 135 mutants: the systematical choices of mutations and over tens of microsecond simulations were impressive. The choices of relevant intramolecular distances (TL-BH etc) also appear reasonable. However, to take advantage of current simulation data, the authors might test if the performances of ML implementations vary depending on the choices of the relevant distances. If say to improve the performance is hard, one can at least test if the performance can be worse by choosing another set of intramolecular distances or even random ones. The reason to do such is to see whether there are correlations between the performance and the choice of the relevant intramolecular distances; if yes, then the performance can be expected to improve in the future once a better or an optimal set of ‘distances’ is obtained. Meanwhile, have the authors try to add features, e.g., dihedrals or energetics from MD to possibly improve the ML performances?

In addition, since the authors suggest that longer or microsecond simulations can possibly improve the performance of the ML implementations, then the authors can test using current MD simulation data set (e.g. 100 ns each trajectory), if a choice of short or non-equilibrated simulations (e.g. 0-50 ns trajectory) or medium stage (e.g. 30-80 ns) impact on the performances, i.e., to make the performances worse. If these impacts are notable, then one has better reasons to believe that longer (>>100 ns) or better equilibrated trajectories has the potential to improve.

(11) Fig 4: the authors well list five groups of distances (A) and (B), but in (C), only show the heatmap for the fully averaged distances. It would be much more informative to show five heatmaps for respective five groups of distances, so that the authors can really point out some of the data structures in the figure to make reasonings, while currently the authors often make statements on structural contributions without referring to any data.

(12) Fig 5: MD wo/w attention. What is ‘attention’, pls explain for readers lack of ML knowledges. In VAE model presented on page 17 “… we also added an attention layer both on the encoder and decoder sides”, pls explain why or what the purpose of adding an attention layer.

(13) Fig 6 on VAE models: Fig 6A, what are the data axis Z[0] and Z[1] represent, or what the matric used to conduct the clustering/K-means? Since different phenotypes scatter in different clusters, no much statistics are provided as well, it is not obvious how one can learn from the clustering in this latent space.

Fig 6B, is it better to show cluster I/II/III data separately (not mixed) to reveal some data structure patterns? It’s also more informative to show for different distance groups (e.g. those five groups in Fig 4AB). Again, the authors should show data explicitly from which they drew conclusions, not just provide conclusions/statements without showing the data.

For example, where are the data that support statements below:

“longer intra-molecular distances for key residues around the TL for LOF/lethal mutants”;

“Cluster II features larger distances between TL residues and GTP compared to the other clusters”;

“Cluster III also features larger distances 321 between TL and GTP for some mutants”;

“cluster III mostly features increased distances for BH-TL pairs”

“Some of the TL-TL distances for the residues that are close to V1094 are also increased for the cluster III”

Fig 6F-H showing “M1079N and E1103G in the clusters III” and “Both M1079N and E1103G lead to disruptions of the hydrophobic pocket although the mechanism may be slightly different”, and concludes “results support the hypothesis that the primary mechanism for GOF phenotypes is via the direct or indirect disruption of the hydrophobic pocket formed by TL residues near the BH as suggested by previous studies (30, 32).”

How many GOFs or how much percentile show the “disruptions of the hydrophobic pocket”? It is not clear whether and how the VAE model and clustering data help to bring the insight on “GOF phenotypes is via the direct or indirect disruption of the hydrophobic pocket formed by TL residues near the BH”.

(14) Conclusions (the last paragraph): I’m wondering about how the “VAE models developed based on MD still could provide mechanistic insights into the potential structural basis of lethal, LOF, and GOF mutants”: are VAE models basically doing the dimension reduction?

Again, I’m also still wondering about where to find the VAE related dataset (or figures) to reach conclusions in the text: (i)“lethal and more serious LOF phenotypes correlate with mutations that directly increase the distance between key TL residues and the GTP in the active site”; (ii) “Weaker LOF phenotypes for the examined mutants appear to be related more to disruptions in the TL-BH interaction”; (iii) “a substantial fraction of GOF phenotypes appear to result largely from a disruption of a hydrophobic pocket near the TL”; (iv)“In most of these GOF mutants, we observed increased BH-TL distances especially for the cases when the mutants located close to BH”.

Reviewer #3: The authors of the manuscript engage is the challenging task of trying to correlate genetic fitness scores with sequence-based machine learning and structural insights from molecular dynamics.

A general issue with the manuscript is that it presents a rather “yeast-centric” perspective. Given the high degree of evolutionary conservation of RNAP active site function across the evolutionary spectrum, the authors do themselves (in my view) a disservice by not emphasizing sufficiently this universality and not taking a broader view in their Introduction and Discussion sections. Inclusion of insights gained from mutations in archaeal and bacterial Trigger Loop mutations would make the manuscript more relevant to a wider readership.

This paper is going to be very challenging for “average” molecular biologists to read (who are the most likely target audience to benefit from the biological take-away messages), so most of the recommendations for improvements given below are aimed at encouraging the authors to explain some of the key biological and computational assumptions and concepts in more detail.

Overall, the authors succeeded only partially in their ambitious goal. This is, however, not meant as a major criticism, but rather reflects the complexity of the topic and the technical shortcomings in obtaining the right biological data sets and limitations of computational resources and approaches. The manuscript should be seen as a brave attempt to tackle a fundamentally important question and gaining insights that are worth publishing.

Specific Recommendations:

Abstract:

Line 45: GOF abbreviation not defined

Author Summary:

Line 58, 59: The sentence concerning effects on the hydrophobic interactions is very vague. Maybe more detail/specific information to sharpen up the argument?

Introduction:

The description in line 68 applies to RNAPs from all cellular organisms. To apparently limit the description to “yeast Pol II” is too narrow. The universality of the NAC in all cellular RNAPs should be emphasized. A new figure, showing the sequence alignment of TL sequences from a variety of archaeal, bacterial, and eukaryotic species, would provide an excellent background to visualize the relationship of residues in yeast RNAP discussed in the manuscript with the degree of evolutionary conservation. Such a sequence alignment is even essential for the reader to get an impression concerning the fundamental question addressed in the manuscript: how do functional TL phenotypes resulting from residue variations manifest themselves on a structural and dynamic level? Some residues are evolutionarily highly conserved, others less so, thus already providing at least partial insight into a potential answer.

Results:

Line 133, 135: What is meant by “indeterminate” phenotype? How is this defined? What are the reasons for combining this with the wildtype data?

In Figure 1, it does not mention what these mutants actually are! A brief description is absolutely essential here. Also, what is the rationale for the different assay conditions? What are they supposed to test? In which parts of RNAPII are the mutations located that are tested in combination with the TL mutants? More information is necessary here.

There is obviously a rather substantial conceptual problem with the phenotypic range data set: averaging results from a range of assays based on potentially mechanistically functions may hide some interesting effects of some of the mutants under certain conditions. The authors mention that this set-up allows them to distinguish between LOF and GOF mutants, but this seems a rather modest achievement compared to the potential richness of mechanistic differences hidden in the primary data under the various assay conditions. Rather than combining the data into an averaged fitness score, the authors should consider running the ML model for each of the experimental conditions independently to see if some of the experimental conditions identify particular mutant properties that gets lost in the average score. Could this averaging effect also explain why extreme outcomes (Line 195, 196) are not predicted reliably by the model because some of the data that would help to predict such outcomes more reliably got “submerged” in the phenotypic score used for the training?

Line 141: “we trained a neural network model” is rather unspecific; even in the main text, the basic architecture (256, 128 and 64 nodes, and 128 and 64 nodes connected to the output layer via an attention layer; as described in Materials and Methods) should be mentioned, especially considering that this manuscript is under consideration for a computational journal. Are there any reasons for choosing these particular architectures? Why was a self-attention framework considered? How do these two architectures compare? Where any other architectures tested? How long was the model trained? Was there any evidence of overfitting and how was this possibility excluded?

Figure 2 provides a great overview of the prediction, but once again I miss an evolutionary sequence alignment at this point. Residues 1077, 1078 and 1080-1088 are predicted to be rather sensitive to mutation, whereas positions 1090-1104 are either insensitive or prone to GOF phenotypes. How does this correlate with the evolutionary conservation? Also, for readers less familiar with TL structure, showing a simple spatial model of the TL structure with at least some of the residue positions marked would be helpful.

Line 214: What is the rationale for attempting double predictions - spell out the reasons for clarity.

Line 221: “This additive prediction is consistent with previous studies” is too enigmatic; not everybody has the time to look up the reference provided. Provide a brief explanation how this data fits previous results.

Line 230, 231: it is a pity that this potentially very powerful prediction has not been tested experimentally yet. A successful outcome would strengthen the value of the model presented in this manuscript considerably!

Line 236: Once again, I consider a brief description of some of the thinking concerning the MD simulation set-up to be essential at this point. Why was a simulation of the whole RNAPII structure (PDB 2E2H) considered to be essential? Longer simulations on a smaller structure only encompassing the elements of the active site could have been a more productive use of computing resources. Given that the authors simulated the whole RNAPII molecule, did they find any evidence that any of the TL mutants had any long-range distance effects on structures outside the active site?

Why was the system only neutralized and not simulated under a more “physiological” (150 mM NaCl) condition? Considering that the NTP is highly charged and that TL-NTP interactions could be affected by ionic interactions, this is a somewhat “courageous” way of setting up the simulations.

Overall, it seems clear that the MD data - despite the huge effort obtaining it - is not very informative. I agree, however, with the authors’ decision to include the data to show the current limitations on these approaches. Obviously, it would have been great if it had worked better (and longer simulations on reduced systems, as suggested earlier, could help).

Line 283: A brief and more general explanation of VAE models, and why this approach was chosen in this instance, would be very helpful. I do not think that biologists reading this manuscript would be able to understand the reasons given in Lines 285- 288.

**Have the authors made all data and (if applicable) computational code underlying the findings in their manuscript fully available?**

Reviewer #1: None

Reviewer #2: None

Reviewer #3: Yes

PLOS authors have the option to publish the peer review history of their article (what does this mean?). If published, this will include your full peer review and any attached files.

Reviewer #1: No

Reviewer #2: No

Reviewer #3: No
---

## [Decision Letter · Decision Letter 1]

30 Nov 2022

Dear Dr. Feig,

Thank you very much for submitting your manuscript "Characterization of RNA Polymerase II Trigger Loop Mutations using Molecular Dynamics Simulations and Machine Learning" for consideration at PLOS Computational Biology.

As with all papers reviewed by the journal, your revised manuscript was reviewed by members of the editorial board and by the three previous reviewers. In light of the reviews (below this email), we would like to invite the resubmission of a significantly-revised version that takes into account the comments of Reviewer #2 and #3..

We cannot make any decision about publication until we have seen the revised manuscript and your response to the reviewers' comments. Your revised manuscript is also likely to be sent to reviewers for further evaluation.

Sincerely,

Guanghong Wei

Academic Editor

PLOS Computational Biology

Nir Ben-Tal

Section Editor

PLOS Computational Biology

Reviewer's Responses to Questions

**Comments to the Authors:**

Reviewer #1: The authors have appropriately addressed all my comments.

Reviewer #2: The authors have made some efforts and partially addressed my concerns. Some of points left for the authors’ further considerations are listed:

Point 2): the authors only modeled the TL closed but not the TL open conformation:

This study is to address for full RNAP elongation function but not only the catalysis step. The TL open to closed transition is highly important for the elongation cycle as it is potentially a rate-limiting step and a dominant conformational change of the RNAP II. Hence, ignoring the TL open conformation is not well justified for the purpose of this study.

Considering that the computational cost is high already for current implementations, the authors should at least make efforts to explicitly address this issue in the ms and clarify the limitation of this work. A better way, since the lab had previously studied the TL open to close transition, as being claimed (Wang et al., 2013, Biophys. J), the authors can consider including some of previous datasets to make improved analyses or perspectives to current work.

Point 3): the request is not asking the authors to run simulations on mutants without TL, but to clarify an interesting issue why “deletion of the TL from different species caused marked reductions in transcription rate” but not fully abolished the transcription activities. The clarifications provided by the authors (with data and the reference) are helpful to be included into the ms.

Point 5): In regard to the fitness scores which serve for the basis of current analyses: the authors provided a ms in accompany with technical details; the authors also quote the reference of Qiu C et al 2016 work and claimed a brief description of the experimental approach in the ms.

Nevertheless, it’s hard to get a brief message on the basic rationale on how to define such a fitness score. If I missed it, authors may consider highlighting the sentences, or providing a couple of sentences to summarize the main messages according to their own understanding. It’s not for technical details but for intuitive understanding, as rationalizing the fitness score serves for somehow a premise of current study.

Point 9): It is true that the prediction does not rely on the spatial information. Since the authors have access to the 3D structure of the RNAP II, they can simply check the structure. Note that the request is not request the authors to re-run anything but to check for their own data that are potentially more interesting than they thought: to find from the 3D structure of RNAP II on whether those mutations show additive properties are those spatially separated in the 3D locations.

Point 10): If the authors tested the different MD data inputs, e.g., TL dihedrals instead of distances, then such tests and results should be included in the ms (indicate where if included already).

Besides, the authors did not get the point of the quest: Using less features is NOT to gain insight but serve for control. If one shows that fewer input features bring worse results, it is equivalent to show current input (with more features) is better than using fewer. Tests on correlation between the #input features with performance score/index can help to determine how much is enough.

Reviewer #3: The manuscript has been clarified in many of the areas that all three referees highlighted. Nevertheless, it remains a difficult and somewhat unsatisfactory manuscript because the author’s response has been rather minimal and several of the criticisms were only addressed in a superficial manner. Especially the descriptions and explanations of key data sets remain problematic for the reader because the authors keep essential information back to publish them separately somewhere else. Full knowledge and understanding of the meaning of these data sets is, however, critical to appreciate the results presented in the current manuscript. Under these circumstances, I wonder why the authors did not wait to publish the experimental data first and possibly afterwards describe the ML-analysis described here.

Combined with the fact that the authors succeeded only partially in their ambitious bioinformatics goals, my overall impression is therefore that the paper, even after the revision, has not really succeeded in providing clear, novel insights into an important biochemical mechanism. Although sophisticated in computational terms, the goal is overambitious and probably not achievable in a satisfactory way using the currently available methodology.

Although I would not have any issues for the paper to be published in its somewhat improved format, I do not see any strong reasons in favor of publication either.

**Have the authors made all data and (if applicable) computational code underlying the findings in their manuscript fully available?**

Reviewer #1: None

Reviewer #2: None

Reviewer #3: **No: **The description of the experimental data set remains very incomplete and this prevents readers from understanding the analyses presented in the current manuscript. This data should be published first.

PLOS authors have the option to publish the peer review history of their article (what does this mean?). If published, this will include your full peer review and any attached files.

Reviewer #1: No

Reviewer #2: No

Reviewer #3: No
---

## [Editor Report · Decision Letter 2]

6 Mar 2023

Dear Dr. Feig,

We are pleased to inform you that your manuscript 'Characterization of RNA Polymerase II Trigger Loop Mutations using Molecular Dynamics Simulations and Machine Learning' has been provisionally accepted for publication in PLOS Computational Biology.

Best regards,

Guanghong Wei

Academic Editor

PLOS Computational Biology

Nir Ben-Tal

Section Editor

PLOS Computational Biology

---

## [Editor Report · Acceptance letter]

17 Mar 2023

PCOMPBIOL-D-22-01243R2 

Characterization of RNA Polymerase II Trigger Loop Mutations using Molecular Dynamics Simulations and Machine Learning

Dear Dr Feig,

I am pleased to inform you that your manuscript has been formally accepted for publication in PLOS Computational Biology. Your manuscript is now with our production department and you will be notified of the publication date in due course.

With kind regards,

Zsofi Zombor
